# Optimistic Online-to-Batch Conversions
# for Accelerated Convergence and Universality

**Yu-Hu Yan, Peng Zhao, Zhi-Hua Zhou**
National Key Laboratory for Novel Software Technology, Nanjing University, China
School of Artificial Intelligence, Nanjing University, China
{yanyh, zhaop, zhouzh}@lamda.nju.edu.cn

## Abstract

In this work, we study offline convex optimization with smooth objectives, where the classical Nesterov's Accelerated Gradient (**NAG**) method achieves the optimal accelerated convergence. Extensive research has aimed to understand **NAG** from various perspectives, and a recent line of work approaches this from the viewpoint of online learning and online-to-batch conversion, emphasizing the role of *optimistic online algorithms* for acceleration. In this work, we contribute to this perspective by proposing novel *optimistic online-to-batch conversions* that incorporate optimism theoretically into the analysis, thereby significantly simplifying the online algorithm design while preserving the optimal convergence rates. Specifically, we demonstrate the effectiveness of our conversions through the following results: *(i)* when combined with simple online gradient descent, our optimistic conversion achieves the optimal accelerated convergence; *(ii)* our conversion also applies to strongly convex objectives, and by leveraging both optimistic online-to-batch conversion and optimistic online algorithms, we achieve the optimal accelerated convergence rate for strongly convex and smooth objectives, for the first time through the lens of online-to-batch conversion; *(iii)* our optimistic conversion can achieve universality to smoothness — applicable to both smooth and non-smooth objectives without requiring knowledge of the smoothness coefficient — and remains efficient as non-universal methods by using only one gradient query in each iteration. Finally, we highlight the effectiveness of our optimistic online-to-batch conversions by a precise correspondence with **NAG**.

## 1 Introduction

Convex optimization [Boyd and Vandenberghe, 2004, Nesterov, 2018] is a core problem in optimization theory. Its simple theoretical foundations and algorithms have made it essential for solving a wide range of real-world problems. Specifically, we focus on the simplest and most standard form:

$$\min_{\mathbf{x} \in \mathcal{X}} f(\mathbf{x}), \tag{1}$$

where $f(\cdot)$ is a convex objective function and $\mathcal{X} \subseteq \mathbb{R}^d$ is the feasible domain. In this paper, we focus on the function-value convergence. Specifically, we aim to minimize the suboptimality gap $f(X_T) - \min_{\mathbf{x} \in \mathcal{X}} f(\mathbf{x})$, where $X_T$ denotes the final output of an algorithm after $T$ iterations.

### 1.1 Accelerated Convergence

When the objective function $f(\cdot)$ is *Lipschitz-continuous*, the simple Gradient Descent (**GD**) is sufficient to achieve the optimal convergence rates, specifically, $\mathcal{O}(T^{-1/2})$ for convex functions and

---

*Correspondence: Peng Zhao <zhaop@lamda.nju.edu.cn>

39th Conference on Neural Information Processing Systems (NeurIPS 2025).

$\mathcal{O}(T^{-1})$ for strongly convex functions [Nemirovski and Yudin, 1983]. However, **GD** is still not fully capable of handling convex smooth optimization.

In convex *smooth* optimization, the pioneering Nesterov's Accelerated Gradient method (**NAG**) [Nesterov, 1983, 2018] achieves an accelerated and optimal convergence rate of $\mathcal{O}(T^{-2})$. **NAG** consists of two steps — a gradient descent step and an extrapolation step, formalized below:

$$\mathbf{y}_t = \Pi_{\mathcal{X}} \left[ \mathbf{z}_{t-1} - \theta_{t-1} \nabla f(\mathbf{z}_{t-1}) \right], \quad \mathbf{z}_t = \mathbf{y}_t + \beta_t(\mathbf{y}_t - \mathbf{y}_{t-1}), \tag{2}$$

where $\Pi_{\mathcal{X}}[\cdot]$ represents the Euclidean projection onto $\mathcal{X}$, $\beta_t$ is the extrapolation parameter and $\theta_t$ denotes the step size. **NAG** with carefully chosen parameters is able to achieve the optimal convergence rate of $\mathcal{O}(T^{-2})$ for convex functions and $\mathcal{O}(\exp(-T/\sqrt{\kappa}))$ for strongly convex functions, respectively, where $\kappa$ represents the condition number [Nesterov, 1983].

## 1.2 Online-to-Batch Conversion for Acceleration

Due to the significant success of **NAG** and acceleration methods, numerous studies have sought to understand them from various perspectives, including ordinary differential equations [Su et al., 2016], game theory [Wang and Abernethy, 2018], and online learning [Cutkosky, 2019], among others. In this work, we focus on understanding **NAG** and acceleration from the perspective of online learning.

From the online learning perspective, a classic approach is through the well-known *online-to-batch conversion (O2B conversion)* [Cesa-Bianchi et al., 2004, Shalev-Shwartz, 2012], which leverages *online learning* algorithms [Hazan, 2016, Orabona, 2019] to address offline optimization tasks. Formally, online learning is a versatile framework that models the interaction between a learner and the environment over time. In the $t$-th round, the learner selects a decision $\mathbf{w}_t$ from a convex compact set $\mathcal{W} \subseteq \mathbb{R}^d$. Simultaneously, the environment adversarially chooses a convex loss function $\ell_t : \mathcal{W} \to \mathbb{R}$. Subsequently, the learner incurs a loss $\ell_t(\mathbf{w}_t)$, receives feedback of the function $\ell_t(\cdot)$, and updates her decision to $\mathbf{w}_{t+1}$. In online learning, the learner aims to minimize the game-theoretic performance measure known as regret [Cesa-Bianchi and Lugosi, 2006], which is formally defined as

$$\text{REG}_T \triangleq \sum_{t=1}^{T} \ell_t(\mathbf{w}_t) - \min_{\mathbf{w} \in \mathcal{W}} \sum_{t=1}^{T} \ell_t(\mathbf{w}). \tag{3}$$

It represents the learner's excess cumulative loss compared with the best fixed comparator in hindsight.

The vanilla O2B conversion is able to achieve the optimal convergence rate for convex Lipschitz-continuous objectives via a black-box use of online learning algorithms. For example, the simple online gradient descent [Zinkevich, 2003] along with the O2B conversion is sufficient to achieve the optimal rate of $\mathcal{O}(T^{-1/2})$ [Nemirovski and Yudin, 1983].

While the vanilla conversion proves effective for Lipschitz objectives, it encounters limitations in convex smooth optimization. To address this challenge, Cutkosky [2019] introduced a key algorithmic modification — evaluating the gradient at the averaged decision along the optimization trajectory rather than at the current decision in each iteration, as the averaged trajectory exhibits greater stability. Therefore, we refer to this enhanced approach as the *stabilized* O2B conversion throughout this paper. To utilize the stability of the averaged trajectory, Cutkosky [2019] incorporated *optimistic online algorithms* [Chiang et al., 2012, Rakhlin and Sridharan, 2013], a powerful technique in modern online learning [Orabona, 2019]. This integration has inspired a growing body of research that leverages the adaptivity of optimistic online learning algorithms to achieve accelerated convergence [Cutkosky, 2019, Kavis et al., 2019, Joulani et al., 2020b, Zhao et al., 2025]. We provide a comprehensive overview of these recent developments in Section 2.2.

## 1.3 Our Contributions

With stabilized O2B conversion, Cutkosky [2019], Kavis et al. [2019], Joulani et al. [2020b] have demonstrated that optimistic algorithms [Nemirovski, 2004] are essential for acceleration [Nesterov, 1983] in convex smooth optimization. Interestingly, in game theory, recent studies reveal a stark separation of the ability of optimism: optimistic methods are necessary for convergence, while non-optimistic methods invariably diverge [Mertikopoulos et al., 2018]. This fundamental difference prompts an important question regarding the role of optimism across broader optimization contexts.

Table 1: Comparison of the stabilized O2B conversion [Cutkosky, 2019] with our optimistic O2B conversion regarding the ability for acceleration. Our optimistic O2B conversion incorporates optimism directly, enabling look-ahead online learning regret, where a non-optimistic algorithm achieves the optimal accelerated rate.

| Property | Stabilized | Ours (Optimistic) |
|---|---|---|
| Online Learning Regret | Standard | Look-ahead |
| Optimism in Conversion | ✗ | ✓ |
| Optimism in Algorithm | ✓ | ✗ |
| Convergence Rate | $\mathcal{O}(T^{-2})$ | $\mathcal{O}(T^{-2})$ |

Motivated by this question, we further investigate the role of optimism in convex smooth optimization. We find that while optimism is essential for acceleration, it does *not* need to be exclusively achieved via online learning algorithms, but can also be realized through the O2B conversion mechanism. Specifically, we propose an *optimistic O2B conversion*, which can *implicitly* incorporate optimistic capabilities in theoretical analysis. Interestingly, this shows that optimism not only plays a crucial role in online learning and game theory, but also in the offline optimization, making it a fundamental principle across different optimization contexts. Our work is inspired by Wang and Abernethy [2018] but takes a fundamentally different approach: we abandon the game-theoretic interpretation and instead develop a novel optimistic O2B conversion framework that directly incorporates optimism into the conversion mechanism. While we arrive at the same intermediate result (in Theorem 1), our framework provides a more direct and unified understanding of how optimism enables acceleration. As a first application, our framework re-derives the algorithm from Wang and Abernethy [2018] (our Algorithm 2), offering new insights on its effectiveness.

Our optimistic conversion can also be extended to strongly convex and smooth objectives, achieving the optimal convergence rate for the first time through O2B conversion. Notably, while optimism remains essential in this challenging setup, it is now distributed between the O2B conversion and the online learning algorithm, and their cooperation makes the optimal convergence attainable.

Furthermore, we consider making our method *universal* [Nesterov, 2015]. An optimization algorithm is called universal if it can automatically achieve the best possible convergence guarantees under smoothness and non-smoothness, without requiring the smoothness parameter. To this end, a natural idea is to leverage the AdaGrad-type step sizes [Duchi et al., 2011] following Kavis et al. [2019]. However, this generally requires querying the gradient oracle twice in each iteration, which is not as efficient as non-universal methods such as **NAG** (2) and is also not efficient enough when the gradient evaluation is costly. The same concern also appears in the work of Kavis et al. [2019]. To this end, building on our first optimistic conversion, we propose a further enhanced optimistic O2B conversion, which is able to achieve the *optimal and universal* convergence rates, while maintaining *only one* gradient query per iteration, making it as efficient as non-universal methods in terms of the gradient query complexity. Our results can also be extended to the stochastic optimization setting with high-probability guarantees, which we defer to Appendix C due to space constraints.

Table 1 presents the comparison between the stabilized O2B conversion of Cutkosky [2019] and our optimistic conversion regarding the ability for acceleration. The core message is — both O2B conversion and online learning algorithms are essential in achieving acceleration. Our optimistic O2B conversion incorporates the optimistic ability theoretically in the analysis and thus enables look-ahead online learning regret, which can liberate the algorithm design. Therefore, even the simple online gradient descent algorithm can achieve the accelerated convergence.

Finally, we highlight the equivalence of the algorithm update trajectory between the optimistic-O2B-induced algorithm and the classical **NAG** (2) in unconstrained settings under specific parameter configurations, offering an interpretation of **NAG** through the lens of optimistic online learning. Conversely, this observation also offers an intuitive explanation for the effectiveness of our optimistic conversion because of its profound connection with **NAG**. Furthermore, we demonstrate that previous works implementing optimistic algorithms [Cutkosky, 2019, Kavis et al., 2019, Joulani et al., 2020b] can be viewed as variants of Polyak's Heavy-Ball method [Polyak, 1964], enhanced with carefully designed corrected gradients.

To conclude, our contributions are mainly threefold, summarized as follows:

- We propose an optimistic O2B conversion framework, which implicitly incorporates optimism in theoretical analysis. Our optimistic conversion re-derives the algorithm from Wang and Abernethy [2018], while from the perspective of online-to-batch conversion, offering new insights on the relationship between the game-theoretic and the O2B-theoretic interpretations.
- Our optimistic conversion extends to strongly convex and smooth objectives. This marks the *first* time that the optimal convergence rate for strongly convex and smooth objectives has been achieved through O2B conversion.
- We further enhance our optimistic conversion to achieve the optimal and universal convergence with only one gradient query per iteration, making it as efficient as non-universal methods in terms of the gradient query complexity, which is validated by the numerical evaluation in Section 5.

**Organization.** The rest of the paper is organized as follows. Section 2 introduces the preliminaries and reviews related work. Section 3 presents our optimistic O2B conversions with accelerated and universal convergence. Section 4 compares O2B methods with classical approaches. Section 5 provides the numerical experiments to evaluate the performance of the proposed methods. Finally, Section 6 concludes the paper. All proofs are provided in appendices.

## 2 Preliminary

In this section, we introduce some preliminary knowledge, including notations, assumptions, and recent advancements in convex smooth optimization via the stabilized O2B conversion.

### 2.1 Notations and Assumptions

In the following, we introduce the notations and assumptions used in this work.

**Notations.** For simplicity, we use $\|\cdot\|$ for $\|\cdot\|_2$ by default and write $a \lesssim b$ or $a = \mathcal{O}(b)$ if there exists a constant $C < \infty$ such that $a/b \leq C$. When there is no ambiguity, we abbreviate $\sum_t$ and $\{\cdot\}_t$ for $\sum_{t=1}^T$ and $\{\cdot\}_{t=1}^T$, respectively. We adopt $\Pi_{\mathcal{X}}[\mathbf{z}] = \arg\min_{\mathbf{x}\in\mathcal{X}} \|\mathbf{z} - \mathbf{x}\|$ to denote the Euclidean projection onto $\mathcal{X}$. Given sequences of O2B conversion weights $\{\alpha_t\}_t$ and decisions of the online algorithm $\{\mathbf{x}_t\}_t$, we use $A_t \triangleq \sum_{s=1}^t \alpha_s$ to represent the weights' summation and define

$$\widetilde{\mathbf{x}}_t \triangleq \frac{1}{A_t}\left(\sum_{s=1}^{t-1} \alpha_s \mathbf{x}_s + \alpha_t \mathbf{x}_{t-1}\right), \text{ and } \bar{\mathbf{x}}_t \triangleq \frac{1}{A_t}\left(\sum_{s=1}^{t-1} \alpha_s \mathbf{x}_s + \alpha_t \mathbf{x}_t\right) \tag{4}$$

to be two kinds of weighted decisions. The two expressions differ only in the decision associated with $\alpha_t$, while all other terms remain the same.

**Assumption 1** (Domain Boundedness)**.** For any $\mathbf{x}, \mathbf{y} \in \mathcal{X} \subseteq \mathbb{R}^d$, the domain satisfies $\|\mathbf{x} - \mathbf{y}\| \leq D$.

**Assumption 2** (Smoothness)**.** The objective function $f(\cdot)$ is $L$-smooth, i.e., $\|\nabla f(\mathbf{x}) - \nabla f(\mathbf{y})\| \leq L\|\mathbf{x} - \mathbf{y}\|$ holds for any $\mathbf{x}, \mathbf{y} \in \mathbb{R}^d$.

Both assumptions are standard in the optimization literature. Specifically, Assumption 1 is common in constrained optimization [Cutkosky, 2019, Kavis et al., 2019, Joulani et al., 2020b]. Note that the boundedness assumption is only required when designing a universal method in Section 3.3. Additionally, Assumption 2 is necessary for the accelerated convergence [Nesterov, 1983, 2018].

### 2.2 (Stabilized) Online-to-Batch Conversion

In this part, we introduce the vanilla online-to-batch conversion [Cesa-Bianchi et al., 2004, Shalev-Shwartz, 2012] and the enhanced stabilized conversion [Cutkosky, 2019].

First, we formalize the vanilla O2B conversion in Algorithm 1, where in each iteration, the learner queries the gradient feedback of $\nabla f(\mathbf{x}_t)$, and feeds $\alpha_t \nabla f(\mathbf{x}_t)$ into the black-box online learning algorithm to generate the next decision $\mathbf{x}_{t+1}$. By doing this, the suboptimality gap of the final decision $\bar{\mathbf{x}}_T$, defined in Eq. (4), satisfies $f(\bar{\mathbf{x}}_T) - f(\mathbf{x}^\star) \leq \frac{1}{A_T}\sum_t\langle\alpha_t\nabla f(\mathbf{x}_t), \mathbf{x}_t - \mathbf{x}^\star\rangle$. The above property is sufficient to attain the optimal convergence rates for Lipschitz-continuous objectives, i.e., $\mathcal{O}(T^{-1/2})$ for convex and $\mathcal{O}(T^{-1})$ for strongly convex functions [Lacoste-Julien et al., 2012],

---

**Algorithm 1** Vanilla/Stabilized Online-to-Batch Conversion

---

**Input:** Online learning algorithm $\mathcal{A}_{\mathsf{OL}}$, weights $\{\alpha_t\}_{t=1}^T$ with $\alpha_t > 0$.

1: **Initialize:** $\mathbf{x}_1 = \mathbf{x}_0 \in \mathcal{X}$
2: **for** $t = 1$ **to** $T$ **do**
3:   Query the gradient feedback $\mathbf{g}_t$, where

$$\mathbf{g}_t = \begin{cases} \nabla f(\mathbf{x}_t), & \text{(vanilla O2B conversion)} \\ \nabla f(\bar{\mathbf{x}}_t), & \text{(stabilized O2B conversion)} \end{cases}$$

4:   Define $\ell_t(\mathbf{x}) \triangleq \langle \alpha_t \mathbf{g}_t, \mathbf{x} \rangle$ as the $t$-th round online function for $\mathcal{A}_{\mathsf{OL}}$
5:   Get $\mathbf{x}_{t+1}$ from $\mathcal{A}_{\mathsf{OL}}(\mathbf{x}_1, \{\ell_s(\cdot)\}_{s=1}^t)$
6: **end for**
7: **Output:** $\bar{\mathbf{x}}_T = \frac{1}{A_T} \sum_{t=1}^T \alpha_t \mathbf{x}_t$

---

matching the known lower bounds [Nemirovski and Yudin, 1983]. Despite its simplicity, vanilla conversion has not yet been shown to yield accelerated rates in smooth convex optimization problems.

To this end, Cutkosky [2019] introduced the more powerful stabilized O2B conversion. The key idea is to query the gradient of the averaged decision $\bar{\mathbf{x}}_t$ rather than $\mathbf{x}_t$ in each iteration. The suboptimality gap of the final decision $\bar{\mathbf{x}}_T$ satisfies $f(\bar{\mathbf{x}}_T) - f(\mathbf{x}^\star) \leq \frac{1}{A_T} \sum_t \langle \alpha_t \nabla f(\bar{\mathbf{x}}_t), \mathbf{x}_t - \mathbf{x}^\star \rangle$. Although the only algorithmic difference between vanilla and stabilized conversion is where the gradient is taken, this is the key reason why stabilized conversion outperforms the vanilla one in terms of acceleration. To exploit the power of the stabilized O2B conversion, Cutkosky [2019] leveraged the optimistic online learning techniques, which follows the two-step update rule below:

$$\mathbf{w}_t = \Pi_{\mathcal{W}}\left[\widehat{\mathbf{w}}_t - \eta_t M_t\right], \quad \widehat{\mathbf{w}}_{t+1} = \Pi_{\mathcal{W}}\left[\widehat{\mathbf{w}}_t - \eta_t \nabla \ell_t(\mathbf{w}_t)\right], \tag{5}$$

where $\eta_t > 0$ is a time-varying step size and $\widehat{\mathbf{w}}_t$ is an intermediate variable. In each iteration, before receiving the gradient feedback, the learner first updates using an optimistic estimate $M_t$ (called an *optimism*) of the future gradient $\nabla \ell_t(\mathbf{w}_t)$ and then updates using the true gradient $\nabla \ell_t(\mathbf{w}_t)$. By doing this, optimistic methods can achieve the following regret bound with adaptivity for any $\mathbf{u} \in \mathcal{W}$:

$$\sum_{t=1}^T \left[\ell_t(\mathbf{w}_t) - \ell_t(\mathbf{u})\right] \leq \mathcal{O}\left(\frac{1}{\eta_T} + \sum_{t=1}^T \eta_t \|\nabla \ell_t(\mathbf{w}_t) - M_t\|_2^2 - \sum_{t=2}^T \frac{1}{\eta_{t-1}} \|\mathbf{w}_t - \mathbf{w}_{t-1}\|_2^2\right), \tag{6}$$

which can result in favorable regret guarantees when the optimistic term $\|\nabla \ell_t(\mathbf{w}_t) - M_t\|$ is small. In the context of offline convex optimization, $\nabla \ell_t(\mathbf{w}_t)$ equals $\alpha_t \nabla f(\bar{\mathbf{x}}_t)$. Subsequently, Cutkosky [2019] chooses $M_t = \alpha_t \nabla f(\bar{\mathbf{x}}_{t-1})$ as the optimism (also known as gradient-variation regret in online learning [Chiang et al., 2012]) such that the optimistic term depends on $\|\bar{\mathbf{x}}_t - \bar{\mathbf{x}}_{t-1}\|$ under smoothness, because the combined decisions $\{\bar{\mathbf{x}}_t\}_t$ is *stable*. To illustrate this, we focus on a key equation proposed therein: $A_{t-1}(\bar{\mathbf{x}}_{t-1} - \bar{\mathbf{x}}_t) = \alpha_t(\bar{\mathbf{x}}_t - \mathbf{x}_t)$. When the domain is bounded by Assumption 1, the variation of $\|\bar{\mathbf{x}}_t - \bar{\mathbf{x}}_{t-1}\| \approx \alpha_t/A_t = 1/t$ using $\alpha_t = 1$. Cutkosky [2019] employed this key stability property to achieve an $\mathcal{O}(T^{-3/2})$ rate in the constrained setup and $\mathcal{O}(\log T \cdot T^{-2})$ for unconstrained optimization.

Consequently, Kavis et al. [2019] leveraged optimistic online learning (5) with optimism $M_t = \alpha_t \nabla f(\widetilde{\mathbf{x}}_t)$, where $\widetilde{\mathbf{x}}_t$ is another form of weighted combination, as defined in Eq. (4). By doing so, the optimistic quantity $\|\nabla \ell_t(\mathbf{w}_t) - M_t\|^2$ is of the same order as $L^2 \|\mathbf{x}_t - \mathbf{x}_{t-1}\|^2$ when $\alpha_t = t$, which therefore can be canceled by the intrinsic negative stability terms in the analysis of optimistic algorithms, as shown in Eq. (6), resulting in the optimal rate of $\mathcal{O}(T^{-2})$. Joulani et al. [2020b] obtained the same optimal convergence rate but with a different optimism configuration of $M_t = \alpha_t \nabla f(\bar{\mathbf{x}}_{t-1})$ and extended the results to composite and variance-reduced optimization. For a comprehensive understanding of the connection between optimistic online learning and acceleration in smooth optimization, one may refer to Zhao [2025].

## 3 Optimistic Online-to-Batch Conversions for Acceleration and Universality

In this section, we introduce our optimistic online-to-batch conversions. Specifically, Section 3.1 proposes the first optimistic conversion for acceleration. Subsequently, Section 3.2 extends our

$$
\text{Stabilized Conversion:} \quad \mathbf{x}_{t-1} \xrightarrow{\mathbf{g}_{t-1}^{\mathsf{S}}} \mathbf{x}_t \xrightarrow{\mathbf{g}_t^{\mathsf{S}}} \mathbf{x}_{t+1}
$$

$$
\text{Optimistic Conversion:} \quad \mathbf{x}_{t-1} \xrightarrow{\mathbf{g}_t^{\mathsf{O}}} \mathbf{x}_t \xrightarrow{\mathbf{g}_{t+1}^{\mathsf{O}}} \mathbf{x}_{t+1}
$$

Figure 1: Comparison of the update between the optimistic and stabilized conversions, where $\mathbf{g}_t^{\mathsf{O}} = \alpha_t \nabla f(\widetilde{\mathbf{x}}_t)$ and $\mathbf{g}_t^{\mathsf{S}} = \alpha_t \nabla f(\bar{\mathbf{x}}_t)$ represent the losses faced by the optimistic and stabilized conversions, and $\mathbf{x} \xrightarrow{\mathbf{g}} \mathbf{y}$ denotes updating from $\mathbf{x}$ to $\mathbf{y}$ using the information $\mathbf{g}$. Compared with the stabilized conversion, ours can update with the information of the *upcoming* losses.

conversion to strongly convex objectives. Finally, Section 3.3 proposes a further enhanced optimistic O2B conversion to achieve universality while maintaining efficiency.

### 3.1 Acceleration for Smooth and Convex Functions

In this part, we propose our first optimistic O2B conversion below, which is general as it only requires the convexity of the objective function. The proof is presented in Appendix B.1.

**Theorem 1.** *If the objective function $f(\cdot)$ is convex, then we have*

$$
A_T \left[ f(\bar{\mathbf{x}}_T) - f(\mathbf{x}^\star) \right] \leq \sum_{t=1}^{T} \langle \alpha_t \nabla f(\widetilde{\mathbf{x}}_t), \mathbf{x}_t - \mathbf{x}^\star \rangle + \sum_{t=1}^{T} \alpha_t \langle \nabla f(\bar{\mathbf{x}}_t) - \nabla f(\widetilde{\mathbf{x}}_t), \mathbf{x}_t - \mathbf{x}_{t-1} \rangle, \quad (7)
$$

*where $\widetilde{\mathbf{x}}_t \triangleq \frac{1}{A_t} \big( \sum_{s=1}^{t-1} \alpha_s \mathbf{x}_s + \alpha_t \mathbf{x}_{t-1} \big)$ and $\bar{\mathbf{x}}_t \triangleq \frac{1}{A_t} \big( \sum_{s=1}^{t-1} \alpha_s \mathbf{x}_s + \alpha_t \mathbf{x}_t \big)$.*

**Comparison to Stablized O2B Conversion [Cutkosky, 2019].** Compared with the stabilized O2B conversion, which exhibits the property of $A_T \left[ f(\bar{\mathbf{x}}_T) - f(\mathbf{x}^\star) \right] \leq \sum_t \langle \alpha_t \nabla f(\bar{\mathbf{x}}_t), \mathbf{x}_t - \mathbf{x}^\star \rangle$, our optimistic O2B conversion in Theorem 1 differs in two aspects:

*(i)* The first term in the right-hand side of Eq. (7) is an algorithm-related *look-ahead* online learning regret that allows a *clairvoyant-type* update because the algorithm can first observe the loss of the $t$-th iteration, i.e., $\alpha_t \nabla f(\widetilde{\mathbf{x}}_t)$, and then updates from $\mathbf{x}_{t-1}$ to $\mathbf{x}_t$. Intuitively, this look-ahead online learning problem is easier to handle than the standard online learning problem. For stabilized conversion, clairvoyant-type updates are forbidden because $\bar{\mathbf{x}}_t$ contains the information of the current decision $\mathbf{x}_t$. Figure 1 illustrates the aforementioned difference.

*(ii)* The second term in Eq. (7) serves as an optimistic quantity in the analysis, which can therefore preserve the potential for acceleration. The difference from the stabilized conversion of Cutkosky [2019] is that our conversion introduces the optimistic term directly in the analysis, whereas the stabilized conversion requires an optimistic algorithm (5) explicitly to achieve the same effect.

Our technical approach differs from the stabilized-O2B-conversion-based methods [Cutkosky, 2019, Kavis et al., 2019, Joulani et al., 2020b] from two aspects: *(i)* Previous works analyze $\alpha_t[f(\bar{\mathbf{x}}_t) - f(\mathbf{x}^\star)]$ as an intermediate quantity, while we focus on $\alpha_t[f(\widetilde{\mathbf{x}}_t) - f(\mathbf{x}^\star)]$; *(ii)* Previous works rely on the key equation $A_{t-1}(\bar{\mathbf{x}}_{t-1} - \bar{\mathbf{x}}_t) = \alpha_t(\bar{\mathbf{x}}_t - \mathbf{x}_t)$, while we employ a novel equation $A_{t-1}(\bar{\mathbf{x}}_{t-1} - \widetilde{\mathbf{x}}_t) = \alpha_t(\widetilde{\mathbf{x}}_t - \mathbf{x}_{t-1})$, which proves crucial for our final guarantee. Interested readers can refer to Appendix A for more useful equations in the O2B conversion.

**Connection to Game-theoretic Interpretation [Wang and Abernethy, 2018].** The authors provided an alternative game-theoretic perspective to understand acceleration in convex optimization. Specifically, they reformulated the convex optimization problem (1) as a two-player Fenchel game: $g(\mathbf{x}, \mathbf{y}) = \langle \mathbf{y}, \mathbf{x} \rangle - f^*(\mathbf{y})$ [Abernethy et al., 2018], where $f^*(\cdot)$ is the Fenchel conjugate of $f(\cdot)$. The performance measure $f(\bar{\mathbf{x}}_T) - f(\mathbf{x}^\star)$ is then expressed as the gap towards the Nash equilibrium: $\sup_{\mathbf{y}} g(\bar{\mathbf{x}}_T, \mathbf{y}) - \inf_{\mathbf{x}} \sup_{\mathbf{y}} g(\mathbf{x}, \mathbf{y})$. Crucially, their analysis incorporated the idea of optimism from online learning into the game-theoretic framework. In contrast, our work reformulates these techniques by removing the game-theoretic interpretation and instead provides a direct O2B conversion perspective. Notably, the technical core of our Theorem 1 closely parallels Theorem 1 in their work, and our framework can also re-derive their Algorithm 2 (our Algorithm 2). We believe that O2B conversion provides a more general and accessible lens for understanding offline optimization compared to game theory. Importantly, our analysis reveals that *optimism is essential for acceleration*.

---

**Algorithm 2** Optimistic Online-to-Batch Conversion with Online Gradient Descent

---

**Input:** Online-to-batch weights $\{\alpha_t\}_{t=1}^T$ and step size $\eta_t$
 1: **Initialize**: $\mathbf{x}_0 \in \mathcal{X}$
 2: **for** $t = 0$ **to** $T - 1$ **do**
 3:      Query $\nabla f(\widetilde{\mathbf{x}}_{t+1})$ where $\widetilde{\mathbf{x}}_{t+1} = \frac{1}{A_{t+1}}\left(\sum_{s=1}^t \alpha_s \mathbf{x}_s + \alpha_{t+1}\mathbf{x}_t\right)$
 4:      Update $\mathbf{x}_{t+1} = \Pi_{\mathcal{X}}\left[\mathbf{x}_t - \eta_t \alpha_{t+1}\nabla f(\widetilde{\mathbf{x}}_{t+1})\right]$
 5: **end for**
 6: **Output**: $\bar{\mathbf{x}}_T$ (in Corollaries 1, 2) or $\widetilde{\mathbf{x}}_T$ (in Theorem 5)

---

Interestingly, optimism also plays a critical role in the game-theoretic approach [Syrgkanis et al., 2015, Zhang et al., 2022], suggesting that optimism may be a fundamental principle across online learning, game theory, and offline optimization.

**Convergence Guarantee.** Consequently, we present a simple yet optimal algorithm for convex smooth optimization. Our algorithm template is summarized in Algorithm 2. In Line 3, unlike the stabilized O2B conversion, we query the gradient $\nabla f(\widetilde{\mathbf{x}}_{t+1})$ rather than $\nabla f(\bar{\mathbf{x}}_t)$ from the oracle. In Line 4, the learner first observes the loss vector of the next round, $\alpha_{t+1}\nabla f(\widetilde{\mathbf{x}}_{t+1})$, and then updates its decision from $\mathbf{x}_t$ to $\mathbf{x}_{t+1}$. This approach differs from classical online learning, where the learner makes her decision first and then observes the loss. This represents the key algorithmic difference between our optimistic O2B conversion and the stabilized conversion. We intentionally leave the step size $\eta_t$, online-to-batch weights $\{\alpha_t\}_{t=1}^T$, and final output unspecified in Algorithm 2 for flexibility.

In the following, we offer the convergence rate of Algorithm 2 and defer the proof to Appendix B.2.

**Corollary 1.** *Under Assumption 2, if $f(\cdot)$ is convex and $\mathcal{X} = \mathbb{R}^d$, Algorithm 2 with weights $\alpha_t = t$ and step size $\eta_t = \frac{1}{4L}$ for any $t \in [T]$ ensures: $f(\bar{\mathbf{x}}_T) - f(\mathbf{x}^\star) \leq \mathcal{O}(L\|\mathbf{x}_0 - \mathbf{x}^\star\|^2/T^2)$.*

Note that the convergence rate scales with the initial distance to the minimizer $\|\mathbf{x}_0 - \mathbf{x}^\star\|$, and we do *not* require the feasible domain to be bounded here, i.e., without Assumption 1. Observe that Algorithm 2 coincides with Algorithm 2 in Wang and Abernethy [2018]; both analyses isolate the same underlying online learning subproblem (Theorem 1). Our contribution is not to introduce a new algorithm, but to showcase how the online-to-batch framework furnishes a fresh route to this effective method and, more importantly, unlocks the new results in Section 3.3.

## 3.2 Acceleration for Smooth and Strongly Convex Functions

In this part, we extend our optimistic O2B conversion in Section 3.1 to strongly convex objectives, where a strongly convex function is formally defined as follows.

**Definition 1** (Strong Convexity). *A function $f(\cdot)$ is $\lambda$-strongly convex over $\mathcal{X}$ if $f(\mathbf{x}) - f(\mathbf{y}) \leq \langle \nabla f(\mathbf{x}), \mathbf{x} - \mathbf{y} \rangle - \frac{\lambda}{2} \cdot \|\mathbf{x} - \mathbf{y}\|^2$ for any $\mathbf{x}, \mathbf{y} \in \mathcal{X}$.*

Since strong convexity is a specialization of convexity, we apply our general stabilized O2B conversion analysis to the convex surrogate $\widehat{f}(\mathbf{x}) \triangleq f(\mathbf{x}) - \frac{\lambda}{2}\|\mathbf{x}\|^2$, and then reconcile the gap between $\widehat{f}(\cdot)$ and the original objective $f(\cdot)$. We present the resulting optimistic O2B conversion for strongly convex objectives in Theorem 2, with the proof deferred to Appendix B.3.

**Theorem 2.** *If the objective function $f(\cdot)$ is $\lambda$-strongly convex (in Definition 1), then we have*

$$A_T\left[f(\bar{\mathbf{x}}_T) - f(\mathbf{x}^\star)\right] \leq \sum_{t=1}^T \alpha_t\left[h_t(\mathbf{x}_t) - h_t(\mathbf{x}^\star)\right] + \sum_{t=1}^T \alpha_t\langle\nabla\widehat{f}(\bar{\mathbf{x}}_t) - \nabla\widehat{f}(\widetilde{\mathbf{x}}_t), \mathbf{x}_t - \mathbf{x}_{t-1}\rangle, \quad (8)$$

*where $\widehat{f}(\cdot) \triangleq f(\cdot) - \frac{\lambda}{2}\|\cdot\|^2$ and $h_t(\cdot) \triangleq \langle\nabla\widehat{f}(\widetilde{\mathbf{x}}_t), \cdot\rangle + \frac{\lambda}{2}\|\cdot\|^2$ is a $\lambda$-strongly convex surrogate.*

Building on Theorem 2, we leverage the one-step variant of optimistic online gradient descent [Joulani et al., 2020a] as the online learning algorithm with carefully designed optimisms to achieve the optimal accelerated convergence. We present the update rule and its guarantee in Theorem 3, with the proof deferred to Appendix B.4.

**Theorem 3.** *Under Assumption 2, suppose the objective $f(\cdot)$ is $\lambda$-strongly convex and $\mathcal{X} = \mathbb{R}^d$. If the algorithm updates as*

$$\mathbf{x}_{t+1} = \mathbf{x}_t - \frac{1}{\lambda A_t}\left(\nabla h_t(\mathbf{x}_t) - M_t + M_{t+1}\right), \quad M_t = \begin{cases} \alpha_1 \nabla \widehat{f}(\widetilde{\mathbf{x}}_1) + \alpha_1 \mathbf{x}_1, & t = 1 \\ \alpha_t \nabla \widehat{f}(\widetilde{\mathbf{x}}_t) + \alpha_t \mathbf{x}_{t-1}, & t \geq 2 \end{cases},$$

*with $\alpha_1 = 1$ and $\alpha_t = \frac{1}{4\sqrt{\kappa}} A_{t-1}$ for $t \geq 2$, then it holds that*

$$f(\bar{\mathbf{x}}_T) - f(\mathbf{x}^\star) \leq \mathcal{O}\left(\lambda \left\|\mathbf{x}_1 - \mathbf{x}^\star\right\|^2 \cdot \exp\left(-\frac{T-1}{1 + 2\sqrt{\kappa}}\right)\right),$$

*where $\kappa \triangleq L/\lambda$ represents the condition number.*

Theorem 2 achieves the optimal convergence rate for strongly convex and smooth objectives, scaling optimally with both the iteration number $T$ and condition number $\kappa$. Notably, this is the *first* time that the optimal convergence rate for strongly convex and smooth objectives has been achieved through online-to-batch conversion.

We note that while optimism remains essential for achieving optimality in strongly convex optimization, the key insight is that the optimistic mechanisms are now distributed between the O2B conversion and the online learning algorithm, making the optimal convergence attainable through their cooperative interaction.

### 3.3 Universality to Smooth and Non-smooth Functions

In this part, we focus on making our optimistic O2B conversion *universal to smoothness* to enhance its robustness while maintaining computational efficiency comparable to non-universal methods. Universality means the method can adapt to both smooth and non-smooth objectives without requiring prior knowledge of the smoothness parameter. This problem has received considerable attention in the literature [Nesterov, 2015, Kavis et al., 2019, Kreisler et al., 2024, Rodomanov et al., 2024, Li and Lan, 2025] as the smoothness parameter is often unknown and challenging to estimate in practice.

To begin with, we demonstrate that Algorithm 2 can be made universal by simply using an AdaGrad-type step size [Duchi et al., 2011], also known as self-confident tuning in the online learning literature [Auer et al., 2002]. The proof is deferred to Appendix B.5.

**Corollary 2.** *Under Assumption 1, if the objective $f(\cdot)$ is convex, Algorithm 2 with weights $\alpha_t = t$ and step sizes*

$$\eta_t = \frac{D}{\sqrt{\sum_{s=1}^{t} \alpha_s^2 \|\nabla f(\bar{\mathbf{x}}_s) - \nabla f(\widetilde{\mathbf{x}}_s)\|^2}}, \tag{9}$$

*guarantees $f(\bar{\mathbf{x}}_T) - f(\mathbf{x}^\star) \leq \mathcal{O}(LD^2/T^2)$ under Assumption 2, and $f(\bar{\mathbf{x}}_T) - f(\mathbf{x}^\star) \leq \mathcal{O}(GD/\sqrt{T})$ when the objective is non-smooth with $\|\nabla f(\cdot)\| \leq G$.*

Although Algorithm 2 with an AdaGrad-type step size is theoretically optimal and universal, its efficiency is limited by requiring two gradient evaluations, $\nabla f(\widetilde{\mathbf{x}}_s)$ and $\nabla f(\bar{\mathbf{x}}_s)$. This is less efficient than non-universal methods such as Theorem 1 and **NAG** (2), which require only one gradient query per iteration. Furthermore, this inefficiency becomes problematic when gradient evaluation is costly, such as in nuclear norm optimization [Ji and Ye, 2009] and mini-batch optimization [Li et al., 2014]. The same limitation appears in the work of Kavis et al. [2019].

In the following, we demonstrate that it is possible to achieve the same optimal convergence rate using *only one* gradient query per iteration through a further improved optimistic O2B conversion. The proof is deferred to Appendix B.6.

**Theorem 4.** *If the objective $f(\cdot)$ is convex, when $\alpha_1 = 1$ and $\alpha_T = 0$, the final term of $A_T\left[f(\widetilde{\mathbf{x}}_T) - f(\mathbf{x}^\star)\right]$ can be bounded by*

$$\sum_{t=1}^{T} \langle \alpha_t \nabla f(\widetilde{\mathbf{x}}_t), \mathbf{x}_t - \mathbf{x}^\star \rangle + \sum_{t=1}^{T-1} \alpha_t \langle \nabla f(\widetilde{\mathbf{x}}_{t+1}) - \nabla f(\widetilde{\mathbf{x}}_t), \mathbf{x}_t - \mathbf{x}_{t-1} \rangle - \sum_{t=2}^{T} A_{t-1} \mathcal{D}_f(\widetilde{\mathbf{x}}_{t-1}, \widetilde{\mathbf{x}}_t), \tag{10}$$

*where $\mathcal{D}_f(\mathbf{x}, \mathbf{y}) \triangleq f(\mathbf{x}) - f(\mathbf{y}) - \langle \nabla f(\mathbf{y}), \mathbf{x} - \mathbf{y} \rangle$ is the Bregman divergence associated with $f(\cdot)$.*

**Remark 1** (Technical Comparison). The main technical difference from previous conversions is that Theorem 4 leverages a novel equation: $A_{t-1}(\widetilde{\mathbf{x}}_t - \widetilde{\mathbf{x}}_{t-1}) = \alpha_t(\mathbf{x}_{t-1} - \widetilde{\mathbf{x}}_t) + \alpha_{t-1}(\mathbf{x}_{t-1} - \mathbf{x}_{t-2})$. This analytical approach completely eliminates $\nabla f(\bar{\mathbf{x}}_t)$ from our derivation, ensuring that the algorithm requires only $\nabla f(\widetilde{\mathbf{x}}_t)$ evaluation at each iteration. Readers can refer to Appendix A for the proof of this equation. ◁

In Theorem 5 below, we present that the above conversion along with online gradient descent can achieve *universal optimal* rates using only one gradient query per iteration. The corresponding proof is deferred to Appendix B.7.

**Theorem 5.** *Under Assumption 1, if the objective $f(\cdot)$ is convex, using weights $\alpha_t = t$ for $t \in [T-1]$, $\alpha_T = 0$, and the step size of*

$$\eta_t = \frac{D}{\sqrt{\sum_{s=1}^t \alpha_s^2 \|\nabla f(\widetilde{\mathbf{x}}_{s+1}) - \nabla f(\widetilde{\mathbf{x}}_s)\|^2}}, \tag{11}$$

*Algorithm 2 enjoys $f(\widetilde{\mathbf{x}}_T) - f(\mathbf{x}^\star) \leq \mathcal{O}(LD^2/T^2)$ under Assumption 2, and $f(\widetilde{\mathbf{x}}_T) - f(\mathbf{x}^\star) \leq \mathcal{O}(GD/\sqrt{T})$ when the objective is non-smooth with $\|\nabla f(\cdot)\| \leq G$.*

Theorem 4 leverages the simple online gradient descent to achieve the universal optimal convergence rates, while requiring only one gradient query per iteration, making it as efficient as non-universal methods such as **NAG** (2) and the method in Corollary 1. Furthermore, we note that our results can be straightforwardly extended to the stochastic optimization setting with high-probability rates, which we defer to Appendix C due to page limits. Below we remark two limitations of Theorem 5.

**Remark 2** (Boundedness Assumption). Theorem 5 requires bounded feasible domains, i.e., only suitable for constrained optimization. We focus on the constrained setup because online learning is naturally suited for handling constraints. Designing universal methods with accelerated convergence in the unconstrained case is highly challenging. Recent work of Kreisler et al. [2024] has made some progress in this direction in the context of "parameter-free" optimization[2] by combining the methods of Kavis et al. [2019] for acceleration and Ivgi et al. [2023] for parameter-freeness. Extending our method to the unconstrained case is highly non-trivial and thus left as an important future direction. ◁

Another limitation of our method is that the weight of the final round $\alpha_T$ must be chosen as $\alpha_T = 0$, which means that the algorithm requires the iteration number $T$ at the beginning.

Finally, we note that achieving the optimal universal rates with strongly convex objectives is still *open* and cannot be directly solved via the universal method proposed in this part. To see this, in Theorem 3, the online-to-batch conversion weight $\alpha_t$ depends on the smoothness $L$. Therefore, in the universal setup where $L$ is unknown, it is more challenging than the convex case because the method needs to estimate the smoothness parameter on the fly, making this problem highly non-trivial.

## 4 Discussions of Conversion-based Methods

In this section, we illuminate the effectiveness of recent O2B conversion methods [Kavis et al., 2019, Joulani et al., 2020b], including ours, by comparing them with classic approaches in convex optimization. Specifically, we first highlight that our algorithm trajectory coincides with **NAG** in Section 4.1. Then we find that stabilized-O2B-conversion-based methods can be interpreted as variants of Polyak's Heavy-Ball in Section 4.2.

### 4.1 Comparing Algorithm 2 with Nesterov's Accelerated Gradient

Our method is algorithmically equivalent to Nesterov's accelerated gradient (**NAG**) in Eq. (2), under certain parameter configurations. To show this, we leverage a result from Wang and Abernethy [2018], with a self-contained proof in Appendix D.

**Proposition 1** (Theorem 4 of Wang and Abernethy [2018]). *If $\mathcal{X} = \mathbb{R}^d$, our Algorithm 2 with step size $\eta_t = \frac{t+1}{t} \cdot \frac{1}{8L}$ is equivalent to **NAG** (2) with $\theta_t = \frac{1}{4L}$ and $\beta_t = \frac{t-1}{t+2}$ such that $\mathbf{y}_t = \bar{\mathbf{x}}_t$ and*

---

[2]The main parameter in our method is the domain diameter $D$, as shown in Eq. (11) in Theorem 5. Therefore, extending our method to unconstrained setup is equivalent to achieving parameter-freeness with acceleration.

$\mathbf{z}_t = \widetilde{\mathbf{x}}_{t+1}$. *Our method can be rewritten as the following equivalent form:*

$$\bar{\mathbf{x}}_t = \bar{\mathbf{x}}_{t-1} + \beta_{t-1}\left(\bar{\mathbf{x}}_{t-1} - \bar{\mathbf{x}}_{t-2}\right) - \frac{1}{4L}\nabla f\left(\bar{\mathbf{x}}_{t-1} + \beta_{t-1}(\bar{\mathbf{x}}_{t-1} - \bar{\mathbf{x}}_{t-2})\right),$$

*which is exactly Nesterov's accelerated gradient method in a one-step update formulation.*

### 4.2 Comparing Previous Methods with Polyak's Heavy-Ball

Kavis et al. [2019], Joulani et al. [2020b] in this thread both adopted the optimistic online learning framework, as shown in Eq. (5), but with slightly different configurations. Specifically, $\nabla \ell_t(\mathbf{w}_t)$ equals $\alpha_t \nabla f(\bar{\mathbf{x}}_t)$ in offline convex optimization. Differently, Kavis et al. [2019] chose the optimism $M_t = \alpha_t \nabla f(\widetilde{\mathbf{x}}_t)$ whereas Joulani et al. [2020b] used $M_t = \alpha_t \nabla f(\bar{\mathbf{x}}_{t-1})$. For clarity, their optimistic updates can be rewritten in a one-step formulation [Joulani et al., 2020a]:

$$\mathbf{x}_{t+1} = \mathbf{x}_t - \eta_t \mathbf{g}_t, \quad \mathbf{g}_t = \begin{cases} -\alpha_t \nabla f(\widetilde{\mathbf{x}}_t) + \alpha_t \nabla f(\bar{\mathbf{x}}_t) + \alpha_{t+1}\nabla f(\widetilde{\mathbf{x}}_{t+1}) & \text{[Kavis et al., 2019]}, \\ -\alpha_t \nabla f(\bar{\mathbf{x}}_{t-1}) + \alpha_t \nabla f(\bar{\mathbf{x}}_t) + \alpha_{t+1}\nabla f(\bar{\mathbf{x}}_t) & \text{[Joulani et al., 2020b]}, \end{cases}$$

where $\mathbf{g}_t$ is a multi-step gradient. Furthermore, due to simple derivations, the above one-step update is equivalent to the following rule in terms of $\{\bar{\mathbf{x}}_t\}_t$:

$$\bar{\mathbf{x}}_t = \bar{\mathbf{x}}_{t-1} + \beta_{t-1}\left(\bar{\mathbf{x}}_{t-1} - \bar{\mathbf{x}}_{t-2}\right) - \eta_{t-1} \cdot \left(\frac{\alpha_t}{A_t}\mathbf{g}_{t-1}\right), \text{ where } \beta_{t-1} = \frac{\alpha_t A_{t-2}}{A_t \alpha_{t-1}}. \tag{12}$$

On the other hand, a classic method in convex optimization named Polyak's Heavy-Ball (**HB**) [Polyak, 1964] equips gradient descent with momentum and updates as

$$\mathbf{z}_t = \mathbf{z}_{t-1} - \beta'_{t-1}(\mathbf{z}_{t-1} - \mathbf{z}_{t-2}) - \eta'_{t-1}\nabla f(\mathbf{z}_{t-1}), \tag{13}$$

where $\beta'_t$ denotes the momentum parameter and $\eta'_t$ is the step size. Comparing Eq. (12) with **HB** (13), we can find that the trajectories of $\{\bar{\mathbf{x}}_t\}_t$ and $\{\mathbf{z}_t\}_t$ are similar except that **HB** uses the gradient of $\nabla f(\mathbf{z}_{t-1})$ while Eq. (12) adopts a corrected gradient of $\frac{\alpha_t}{A_t}\mathbf{g}_{t-1}$. Note that although **HB** itself *cannot* achieve acceleration for general convex smooth objectives, its variants with corrected gradients can do this for (strongly) convex and smooth objectives [Wei and Chen, 2025]. Therefore, previous stabilized-conversion-based methods can be treated as variants of **HB** with corrected gradients, which might explain their success in achieving acceleration.

## 5 Experiments

In this section, we conduct numerical experiments to validate the effectiveness of our proposed methods. We evaluate our methods in the squared loss minimization and logistic regression tasks across multiple LIBSVM datasets under both non-universal and universal settings, comparing against classic methods including **NAG**, gradient descent, UniXGrad [Kavis et al., 2019], and the method in Joulani et al. [2020b]. The results demonstrate that our method achieves comparable or superior convergence performance while maintaining competitive computational efficiency. **Detailed setup descriptions and experimental results can be found in Appendix E.**

## 6 Conclusion

In this paper, we focus on convex smooth optimization and study the role of optimism for achieving acceleration. Previous state-of-the-art methods rely on the stabilized O2B conversion and achieve the ability of acceleration via optimistic online learning algorithms. In this work, we propose optimistic online-to-batch conversions that introduce optimism implicitly in the analysis, allowing acceleration using the simple online gradient descent. Our optimistic online-to-batch conversion can also be extended to the strongly convex case with the optimal convergence therein. Furthermore, we consider making our method universal to smoothness and introduce an improved optimistic online-to-batch conversion method that only requires one gradient query per iteration, making it as efficient as non-universal methods, while maintaining the optimal convergence rates. We also conduct the numerical experiments to evaluate the performance of the proposed methods.

Two directions are worth future exploration. The first is to extend our method to unconstrained domains, which is highly non-trivial and challenging. Recent advances in parameter-free stochastic optimization [Ivgi et al., 2023] or using ensemble ideas for a bounded-to-unbounded reduction [Luo et al., 2022] might prove useful. The second direction is investigate the power of our optimistic online-to-batch conversions in more practical tasks, such as real-world deep learning training, following the recent advance of Defazio et al. [2024].

## Acknowledgements

This research was supported by NSFC (62361146852, 62176117). Peng Zhao would like to thank Jun-Kun Wang for the helpful discussions. The authors also thank the reviewers for their valuable suggestions, which helped improve this paper.

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

# A    Useful Equations in Online-to-Batch Conversion

In this section, we provide some useful equations in the online-to-batch conversion. We define $A_0 = 0$ for completeness and assume $\alpha_1 = 1$ (thus $\widetilde{\mathbf{x}}_1 = \mathbf{x}_0$ and $\bar{\mathbf{x}}_1 = \mathbf{x}_1$) for simplicity.

$$A_{t-1}(\bar{\mathbf{x}}_{t-1} - \bar{\mathbf{x}}_t) = \alpha_t(\bar{\mathbf{x}}_t - \mathbf{x}_t), \tag{14}$$

$$A_t(\widetilde{\mathbf{x}}_t - \bar{\mathbf{x}}_t) = \alpha_t(\mathbf{x}_{t-1} - \mathbf{x}_t), \tag{15}$$

$$A_{t-1}(\bar{\mathbf{x}}_{t-1} - \widetilde{\mathbf{x}}_t) = \alpha_t(\widetilde{\mathbf{x}}_t - \mathbf{x}_{t-1}), \tag{16}$$

$$A_{t-1}(\widetilde{\mathbf{x}}_t - \widetilde{\mathbf{x}}_{t-1}) = \alpha_t(\mathbf{x}_{t-1} - \widetilde{\mathbf{x}}_t) + \alpha_{t-1}(\mathbf{x}_{t-1} - \mathbf{x}_{t-2}). \tag{17}$$

Eq. (14) is the key equation in the analysis of Cutkosky [2019], Eq. (15) is essential for the analysis of Kavis et al. [2019], and the last two equations are the key equations in our analysis. For boundary cases, Eq. (14)-Eq. (16) holds from $t = 1$ and Eq. (17) holds from $t = 2$.

In the following, we provide the corresponding proofs of Eq. (14)-Eq. (17).

*Proof of Eq. (16).* For $t = 1$, it holds trivially as $A_0(\bar{\mathbf{x}}_0 - \widetilde{\mathbf{x}}_1) = \alpha_1(\widetilde{\mathbf{x}}_1 - \mathbf{x}_0) = 0$. For $t \geq 2$,

$$
\begin{aligned}
A_{t-1}(\bar{\mathbf{x}}_{t-1} - \widetilde{\mathbf{x}}_t) &= A_{t-1}\left(\frac{1}{A_{t-1}}\left(\sum_{s=1}^{t-1}\alpha_s\mathbf{x}_s\right) - \frac{1}{A_t}\left(\sum_{s=1}^{t-1}\alpha_s\mathbf{x}_s + \alpha_t\mathbf{x}_{t-1}\right)\right) \\
&= \sum_{s=1}^{t-1}\alpha_s\mathbf{x}_s - \frac{A_{t-1}}{A_t}\left(\sum_{s=1}^{t-1}\alpha_s\mathbf{x}_s + \alpha_t\mathbf{x}_{t-1}\right) \\
&= \left(\sum_{s=1}^{t-1}\alpha_s\mathbf{x}_s + \alpha_t\mathbf{x}_{t-1}\right) - \frac{A_{t-1}}{A_t}\left(\sum_{s=1}^{t-1}\alpha_s\mathbf{x}_s + \alpha_t\mathbf{x}_{t-1}\right) - \alpha_t\mathbf{x}_{t-1} \\
&= \frac{\alpha_t}{A_t}\left(\sum_{s=1}^{t-1}\alpha_s\mathbf{x}_s + \alpha_t\mathbf{x}_{t-1}\right) - \alpha_t\mathbf{x}_{t-1} = \alpha_t(\widetilde{\mathbf{x}}_t - \mathbf{x}_{t-1}),
\end{aligned}
$$

which finishes the proof.    $\square$

*Proof of Eq. (17).* For $t = 2$, it holds trivially as $A_1(\widetilde{\mathbf{x}}_2 - \widetilde{\mathbf{x}}_1) = \alpha_2(\mathbf{x}_1 - \widetilde{\mathbf{x}}_2) + \alpha_1(\mathbf{x}_1 - \mathbf{x}_0)$. For $t > 2$, this can be proved by directly using the definitions of $\widetilde{\mathbf{x}}_t$ and $\bar{\mathbf{x}}_t$ in (4):

$$
\begin{aligned}
A_{t-1}(\widetilde{\mathbf{x}}_t - \widetilde{\mathbf{x}}_{t-1}) &= A_{t-1}\left(\frac{1}{A_t}\left(\sum_{s=1}^{t-1}\alpha_s\mathbf{x}_s + \alpha_t\mathbf{x}_{t-1}\right) - \frac{1}{A_{t-1}}\left(\sum_{s=1}^{t-2}\alpha_s\mathbf{x}_s + \alpha_{t-1}\mathbf{x}_{t-2}\right)\right) \\
&= \frac{A_{t-1}}{A_t}\left(\sum_{s=1}^{t-1}\alpha_s\mathbf{x}_s + \alpha_t\mathbf{x}_{t-1}\right) - \left(\sum_{s=1}^{t-2}\alpha_s\mathbf{x}_s + \alpha_{t-1}\mathbf{x}_{t-2}\right) \\
&= \left(\frac{A_{t-1}}{A_t} - 1\right)\left(\sum_{s=1}^{t-1}\alpha_s\mathbf{x}_s + \alpha_t\mathbf{x}_{t-1}\right) + (\alpha_{t-1}\mathbf{x}_{t-1} - \alpha_{t-1}\mathbf{x}_{t-2} + \alpha_t\mathbf{x}_{t-1}) \\
&= -\alpha_t\widetilde{\mathbf{x}}_t + (\alpha_{t-1}\mathbf{x}_{t-1} - \alpha_{t-1}\mathbf{x}_{t-2} + \alpha_t\mathbf{x}_{t-1}) \\
&= \alpha_t(\mathbf{x}_{t-1} - \widetilde{\mathbf{x}}_t) + \alpha_{t-1}(\mathbf{x}_{t-1} - \mathbf{x}_{t-2}).
\end{aligned}
$$

Besides, it can also be proved by combining Eq. (16) and Eq. (15):

$$
\begin{aligned}
A_{t-1}(\widetilde{\mathbf{x}}_t - \widetilde{\mathbf{x}}_{t-1}) &= A_{t-1}(\widetilde{\mathbf{x}}_t - \bar{\mathbf{x}}_{t-1}) + A_{t-1}(\bar{\mathbf{x}}_{t-1} - \widetilde{\mathbf{x}}_{t-1}) \\
&= \alpha_t(\mathbf{x}_{t-1} - \widetilde{\mathbf{x}}_t) + \alpha_{t-1}(\mathbf{x}_{t-1} - \mathbf{x}_{t-2}),
\end{aligned}
$$

which finishes the proof.    $\square$

# B    Proof for Section 3

In this section, we provide the omitted proofs for Section 3, including Theorem 1, Corollary 1, Proposition 1, and Theorem 2.

## B.1 Proofs of Theorem 1

*Proof.* We start with the analysis with the following quantity:

$$\alpha_t \left[ f(\widetilde{\mathbf{x}}_t) - f(\mathbf{x}^\star) \right] \le \langle \alpha_t \nabla f(\widetilde{\mathbf{x}}_t), \mathbf{x}_{t-1} - \mathbf{x}^\star \rangle + \langle \alpha_t \nabla f(\widetilde{\mathbf{x}}_t), \widetilde{\mathbf{x}}_t - \mathbf{x}_{t-1} \rangle,$$

using the convexity of $f(\cdot)$. Later, we analyze the second term above. Specifically, using the definition of $\widetilde{\mathbf{x}}_t$ and $\bar{\mathbf{x}}_t$ defined in Eq. (4), we have $A_{t-1} \left( \bar{\mathbf{x}}_{t-1} - \widetilde{\mathbf{x}}_t \right) = \alpha_t (\widetilde{\mathbf{x}}_t - \mathbf{x}_{t-1})$. A detailed derivation of this equation is given in Appendix A. Therefore, using the convexity of $f(\cdot)$: $f(\widetilde{\mathbf{x}}_t) - f(\bar{\mathbf{x}}_{t-1}) \le \langle \nabla f(\widetilde{\mathbf{x}}_t), \widetilde{\mathbf{x}}_t - \bar{\mathbf{x}}_{t-1} \rangle$, it holds that

$$\alpha_t \left[ f(\widetilde{\mathbf{x}}_t) - f(\mathbf{x}^\star) \right] \le \langle \alpha_t \nabla f(\widetilde{\mathbf{x}}_t), \mathbf{x}_{t-1} - \mathbf{x}^\star \rangle + A_{t-1} \left[ f(\bar{\mathbf{x}}_{t-1}) - f(\widetilde{\mathbf{x}}_t) \right].$$

Summing over $t \in [T]$, we have

$$- A_T f(\mathbf{x}^\star) \le \sum_{t=1}^{T} \langle \alpha_t \nabla f(\widetilde{\mathbf{x}}_t), \mathbf{x}_{t-1} - \mathbf{x}^\star \rangle + \sum_{t=1}^{T} \left[ A_{t-1} f(\bar{\mathbf{x}}_{t-1}) - A_t f(\widetilde{\mathbf{x}}_t) \right]$$

$$= \sum_{t=1}^{T} \langle \alpha_t \nabla f(\widetilde{\mathbf{x}}_t), \mathbf{x}_{t-1} - \mathbf{x}^\star \rangle + \sum_{t=1}^{T} \left[ A_{t-1} f(\bar{\mathbf{x}}_{t-1}) - A_t f(\bar{\mathbf{x}}_t) \right] + \sum_{t=1}^{T} \left[ A_t f(\bar{\mathbf{x}}_t) - A_t f(\widetilde{\mathbf{x}}_t) \right]$$

$$= \sum_{t=1}^{T} \langle \alpha_t \nabla f(\widetilde{\mathbf{x}}_t), \mathbf{x}_{t-1} - \mathbf{x}^\star \rangle - A_T f(\bar{\mathbf{x}}_T) + \sum_{t=1}^{T} \left[ A_t f(\bar{\mathbf{x}}_t) - A_t f(\widetilde{\mathbf{x}}_t) \right].$$

Using the convexity as $f(\bar{\mathbf{x}}_t) - f(\widetilde{\mathbf{x}}_t) \le \langle \nabla f(\bar{\mathbf{x}}_t), \bar{\mathbf{x}}_t - \widetilde{\mathbf{x}}_t \rangle$ again, it is equivalent to

$$A_T \left[ f(\bar{\mathbf{x}}_T) - f(\mathbf{x}^\star) \right] \le \sum_{t=1}^{T} \langle \alpha_t \nabla f(\widetilde{\mathbf{x}}_t), \mathbf{x}_{t-1} - \mathbf{x}^\star \rangle + \sum_{t=1}^{T} \langle A_t \nabla f(\bar{\mathbf{x}}_t), \bar{\mathbf{x}}_t - \widetilde{\mathbf{x}}_t \rangle$$

$$= \sum_{t=1}^{T} \langle \alpha_t \nabla f(\widetilde{\mathbf{x}}_t), \mathbf{x}_{t-1} - \mathbf{x}^\star \rangle + \sum_{t=1}^{T} \langle \alpha_t \nabla f(\bar{\mathbf{x}}_t), \mathbf{x}_t - \mathbf{x}_{t-1} \rangle$$

$$= \sum_{t=1}^{T} \langle \alpha_t \nabla f(\widetilde{\mathbf{x}}_t), \mathbf{x}_t - \mathbf{x}^\star \rangle + \sum_{t=1}^{T} \langle \alpha_t \nabla f(\bar{\mathbf{x}}_t) - \nabla f(\widetilde{\mathbf{x}}_t), \mathbf{x}_t - \mathbf{x}_{t-1} \rangle,$$

where the second step is due to $A_t(\widetilde{\mathbf{x}}_t - \bar{\mathbf{x}}_t) = \alpha_t(\mathbf{x}_{t-1} - \mathbf{x}_t)$ from the definitions of $\widetilde{\mathbf{x}}_t$ and $\bar{\mathbf{x}}_t$. The proof is completed. □

## B.2 Proof of Corollary 1

Before providing the proof, we list a useful lemma for online mirror descent.

**Lemma 1** (Lemma 4 of Zhao et al. [2024]). *Let $\mathcal{X}$ be a convex set in a Banach space, and $f : \mathcal{X} \mapsto \mathbb{R}$ be a closed proper convex function on $\mathcal{X}$. Given a convex regularizer $\psi : \mathcal{X} \mapsto \mathbb{R}$, with its corresponding Bregman divergence denoted by $\mathcal{D}_\psi(\cdot, \cdot)$. Then any update of the form*

$$\mathbf{x}_t = \arg\min_{\mathbf{x} \in \mathcal{X}} \left\{ h(\mathbf{x}) + \mathcal{D}_\psi(\mathbf{x}, \mathbf{x}_{t-1}) \right\}$$

*satisfies the following inequality for any $\mathbf{u} \in \mathcal{X}$,*

$$h(\mathbf{x}_t) - h(\mathbf{u}) \le \mathcal{D}_\psi(\mathbf{u}, \mathbf{x}_{t-1}) - \mathcal{D}_\psi(\mathbf{u}, \mathbf{x}_t) - \mathcal{D}_\psi(\mathbf{x}_t, \mathbf{x}_{t-1}).$$

*Proof of Corollary 1.* We start from the intermediate result of Theorem 1:

$$A_T \left[ f(\bar{\mathbf{x}}_T) - f(\mathbf{x}^\star) \right] \le \underbrace{\sum_{t=1}^{T} \langle \alpha_t \nabla f(\widetilde{\mathbf{x}}_t), \mathbf{x}_t - \mathbf{x}^\star \rangle}_{\text{TERM (A)}} + \underbrace{\sum_{t=1}^{T} \alpha_t \langle \nabla f(\bar{\mathbf{x}}_t) - \nabla f(\widetilde{\mathbf{x}}_t), \mathbf{x}_t - \mathbf{x}_{t-1} \rangle}_{\text{TERM (B)}}.$$

For TERM (A), using the update rule of $\mathbf{x}_{t+1} = \Pi_{\mathcal{X}} \left[ \mathbf{x}_t - \eta_t \alpha_{t+1} \nabla f(\widetilde{\mathbf{x}}_{t+1}) \right]$ with step size $\eta_t = \eta = \frac{1}{4L}$ as shown in Algorithm 2, via Lemma 1, we obtain

$$\text{TERM (A)} \le \sum_{t=1}^{T} \frac{1}{2\eta_{t-1}} \left( \|\mathbf{x}_{t-1} - \mathbf{x}^\star\|^2 - \|\mathbf{x}_t - \mathbf{x}^\star\|^2 \right) - \sum_{t=1}^{T} \frac{1}{2\eta_{t-1}} \|\mathbf{x}_t - \mathbf{x}_{t-1}\|^2$$

$$\leq \frac{1}{2\eta} \sum_{t=1}^{T} \left( \|\mathbf{x}_{t-1} - \mathbf{x}^\star\|^2 - \|\mathbf{x}_t - \mathbf{x}^\star\|^2 \right) - \frac{1}{2\eta} \sum_{t=1}^{T} \|\mathbf{x}_t - \mathbf{x}_{t-1}\|^2$$

$$\leq \frac{\|\mathbf{x}_0 - \mathbf{x}^\star\|^2}{2\eta} - \frac{1}{2\eta} \sum_{t=1}^{T} \|\mathbf{x}_t - \mathbf{x}_{t-1}\|^2.$$

For TERM (B), using the smoothness assumption and the observation of $A_t(\widetilde{\mathbf{x}}_t - \bar{\mathbf{x}}_t) = \alpha_t(\mathbf{x}_{t-1} - \mathbf{x}_t)$ as shown in Eq. (15), we have

$$\text{TERM (B)} \leq L \sum_{t=1}^{T} \alpha_t \|\bar{\mathbf{x}}_t - \widetilde{\mathbf{x}}_t\| \|\mathbf{x}_t - \mathbf{x}_{t-1}\| = L \sum_{t=1}^{T} \frac{\alpha_t^2}{A_t} \|\mathbf{x}_t - \mathbf{x}_{t-1}\|^2.$$

Combining the two terms, we have

$$A_T \left[ f(\bar{\mathbf{x}}_T) - f(\mathbf{x}^\star) \right] \leq \frac{\|\mathbf{x}_0 - \mathbf{x}^\star\|^2}{2\eta} + \sum_{t=1}^{T} \left( \frac{L\alpha_t^2}{A_t} - \frac{1}{2\eta} \right) \|\mathbf{x}_t - \mathbf{x}_{t-1}\|^2 \leq 2L\|\mathbf{x}_0 - \mathbf{x}^\star\|^2.$$

Finally, using $A_T = \Theta(T^2)$ completes the proof. □

## B.3 Proof of Theorem 2

*Proof.* We first define $\widehat{f}(\mathbf{x}) \triangleq f(\mathbf{x}) - \frac{\lambda}{2}\|\mathbf{x}\|^2$. Since $f(\cdot)$ is $\lambda$-strongly convex, $\widehat{f}(\cdot)$ is convex. As a result, we directly reuse Theorem 1 for convex objectives:

$$A_T \left[ \widehat{f}(\bar{\mathbf{x}}_T) - \widehat{f}(\mathbf{x}^\star) \right] \leq \sum_{t=1}^{T} \langle \alpha_t \nabla \widehat{f}(\widetilde{\mathbf{x}}_t), \mathbf{x}_t - \mathbf{x}^\star \rangle + \sum_{t=1}^{T} \alpha_t \langle \nabla \widehat{f}(\bar{\mathbf{x}}_t) - \nabla \widehat{f}(\widetilde{\mathbf{x}}_t), \mathbf{x}_t - \mathbf{x}_{t-1} \rangle.$$

The left-hand side of the above inequality can be expanded as

$$A_T \left[ \widehat{f}(\bar{\mathbf{x}}_T) - \widehat{f}(\mathbf{x}^\star) \right] = A_T \left[ f(\bar{\mathbf{x}}_T) - f(\mathbf{x}^\star) \right] - \frac{\lambda}{2} A_T \|\bar{\mathbf{x}}_T\|^2 + \frac{\lambda}{2} A_T \|\mathbf{x}^\star\|^2.$$

Moving the last two terms into the right-hand side, we have

$$A_T \left[ f(\bar{\mathbf{x}}_T) - f(\mathbf{x}^\star) \right] \leq \underbrace{\sum_{t=1}^{T} \langle \alpha_t \nabla \widehat{f}(\widetilde{\mathbf{x}}_t), \mathbf{x}_t - \mathbf{x}^\star \rangle + \frac{\lambda}{2} A_T \|\bar{\mathbf{x}}_T\|^2 - \frac{\lambda}{2} A_T \|\mathbf{x}^\star\|^2}_{\text{REG}_T}$$

$$+ \underbrace{\sum_{t=1}^{T} \alpha_t \langle \nabla \widehat{f}(\bar{\mathbf{x}}_t) - \nabla \widehat{f}(\widetilde{\mathbf{x}}_t), \mathbf{x}_t - \mathbf{x}_{t-1} \rangle}_{\text{ADAPTIVITY}}.$$

In the following, we focus on the REG$_T$ term. Specifically, because

$$A_T \|\bar{\mathbf{x}}_T\|^2 = \frac{1}{A_T} \left\| \sum_{t=1}^{T} \alpha_t \mathbf{x}_t \right\|^2 \leq \frac{1}{A_T} \left( \sum_{t=1}^{T} \alpha_t \|\mathbf{x}_t\| \right)^2 = \frac{1}{A_T} \left( \sum_{t=1}^{T} \sqrt{\alpha_t} \cdot \sqrt{\alpha_t} \|\mathbf{x}_t\| \right)^2 \leq \sum_{t=1}^{T} \alpha_t \|\mathbf{x}_t\|^2,$$

where the last step uses the Cauchy-Schwarz inequality, this term can be rewritten as

$$\text{REG}_T \leq \sum_{t=1}^{T} \alpha_t \left( \langle \nabla \widehat{f}(\widetilde{\mathbf{x}}_t), \mathbf{x}_t \rangle + \frac{\lambda}{2} \|\mathbf{x}_t\|^2 \right) - \sum_{t=1}^{T} \alpha_t \left( \langle \nabla \widehat{f}(\widetilde{\mathbf{x}}_t), \mathbf{x}^\star \rangle + \frac{\lambda}{2} \|\mathbf{x}^\star\|^2 \right)$$

$$= \sum_{t=1}^{T} \alpha_t \left[ h_t(\mathbf{x}_t) - h_t(\mathbf{x}^\star) \right],$$

where the last step defines the surrogate loss function of

$$h_t(\mathbf{x}) \triangleq \langle \nabla \widehat{f}(\widetilde{\mathbf{x}}_t), \mathbf{x} \rangle + \frac{\lambda}{2} \|\mathbf{x}\|^2,$$

finishing the proof. □

## B.4 Proof of Theorem 3

For strongly convex objective, we use the one-step variant of optimistic online gradient descent as the optimization algorithm, whose guarantee is presented below.

**Lemma 2.** *The one-step optimistic online gradient descent algorithm updating as*

$$\mathbf{x}_{t+1} = \mathbf{x}_t - \eta_t \left( \nabla f_t(\mathbf{x}_t) - M_t + M_{t+1} \right), \tag{18}$$

*ensures that*

$$\sum_{t=1}^{T} \langle \nabla f_t(\mathbf{x}_t), \mathbf{x}_t - \mathbf{x}^\star \rangle \leq \sum_{t=1}^{T} \left( \langle \nabla f_t(\mathbf{x}_t) - M_t, \mathbf{x}_t - \mathbf{x}_{t+1} \rangle - \frac{1}{2\eta_t} \|\mathbf{x}_t - \mathbf{x}_{t+1}\|^2 \right)$$

$$+ \sum_{t=1}^{T} \left( \frac{1}{2\eta_t} - \frac{1}{2\eta_{t-1}} \right) \|\mathbf{x}_t - \mathbf{x}^\star\|^2 + \frac{1}{2\eta_1} \|\mathbf{x}_1 - \mathbf{x}^\star\|^2.$$

*Proof of Theorem 3.* For the regret part of $\sum_{t=1}^{T} \alpha_t \left[ h_t(\mathbf{x}_t) - h_t(\mathbf{x}^\star) \right]$ in Theorem 2, using $\lambda$-strong convexity, we have

$$\sum_{t=1}^{T} \alpha_t \left[ h_t(\mathbf{x}_t) - h_t(\mathbf{x}^\star) \right] \leq \sum_{t=1}^{T} \langle \alpha_t \nabla h_t(\mathbf{x}_t), \mathbf{x}_t - \mathbf{x}^\star \rangle - \frac{\lambda}{2} \sum_{t=1}^{T} \alpha_t \|\mathbf{x}_t - \mathbf{x}^\star\|^2.$$

Due to Lemma 2, the linearized term above can be controlled as

$$\sum_{t=1}^{T} \langle \alpha_t \nabla h_t(\mathbf{x}_t), \mathbf{x}_t - \mathbf{x}^\star \rangle \leq \underbrace{\sum_{t=1}^{T} \left( \langle \alpha_t \nabla h_t(\mathbf{x}_t) - M_t, \mathbf{x}_t - \mathbf{x}_{t+1} \rangle - \frac{C_1}{2\eta_t} \|\mathbf{x}_t - \mathbf{x}_{t+1}\|^2 \right)}_{\text{TERM (A)}}$$

$$+ \underbrace{\sum_{t=1}^{T} \left( \frac{1}{2\eta_t} - \frac{1}{2\eta_{t-1}} \right) \|\mathbf{x}_t - \mathbf{x}^\star\|^2 + \frac{1}{2\eta_1} \|\mathbf{x}_1 - \mathbf{x}^\star\|^2}_{\text{TERM (B)}} - \underbrace{\sum_{t=1}^{T} \frac{1 - C_1}{2\eta_t} \|\mathbf{x}_t - \mathbf{x}_{t+1}\|^2}_{\text{TERM (C)}},$$

where $C_1 \in (0, 1)$ is an arbitrary constant in the analysis that will be specified later. By choosing the optimism as $M_t = \alpha_t \nabla \widehat{f}(\widetilde{\mathbf{x}}_t) + \alpha_t \mathbf{x}_{t-1}$ for $t \geq 2$ and $M_1 = \alpha_1 \nabla \widehat{f}(\widetilde{\mathbf{x}}_1) + \alpha_1 \mathbf{x}_1$,[3] we have

$$\text{TERM (A)} \leq \sum_{t=1}^{T} \frac{\eta_t}{2C_0} \|\alpha_t \nabla h_t(\mathbf{x}_t) - M_t\|^2 + \sum_{t=1}^{T} \frac{2C_0 - C_1}{2\eta_t} \|\mathbf{x}_t - \mathbf{x}_{t+1}\|^2$$

$$= \sum_{t=2}^{T} \frac{\lambda^2 \eta_t \alpha_t^2}{2C_0} \|\mathbf{x}_t - \mathbf{x}_{t-1}\|^2 + \sum_{t=1}^{T} \frac{2C_0 - C_1}{2\eta_t} \|\mathbf{x}_t - \mathbf{x}_{t+1}\|^2$$

$$\leq \sum_{t=2}^{T} \left( \frac{\lambda^2 \eta_t \alpha_t^2}{2C_0} + \frac{2C_0 - C_1}{2\eta_{t-1}} \right) \|\mathbf{x}_t - \mathbf{x}_{t-1}\|^2.$$

Combining all terms, we obtain

$$\sum_{t=1}^{T} \alpha_t \left[ h_t(\mathbf{x}_t) - h_t(\mathbf{x}^\star) \right] \leq \sum_{t=2}^{T} \left( \frac{\lambda^2 \eta_t \alpha_t^2}{2C_0} + \frac{2C_0 - C_1}{2\eta_{t-1}} \right) \|\mathbf{x}_t - \mathbf{x}_{t-1}\|^2$$

$$+ \sum_{t=1}^{T} \left( \frac{1}{2\eta_t} - \frac{1}{2\eta_{t-1}} - \frac{\lambda \alpha_t}{2} \right) \|\mathbf{x}_t - \mathbf{x}^\star\|^2 + \frac{1}{2\eta_1} \|\mathbf{x}_1 - \mathbf{x}^\star\|^2 - \sum_{t=1}^{T} \frac{1 - C_1}{2\eta_t} \|\mathbf{x}_t - \mathbf{x}_{t+1}\|^2$$

$$= \sum_{t=2}^{T} \left( \frac{\lambda \alpha_t^2}{2C_0 A_t} + \frac{\lambda A_{t-1}(2C_0 - C_1)}{2} \right) \|\mathbf{x}_t - \mathbf{x}_{t-1}\|^2 + \frac{\lambda \alpha_1}{2} \|\mathbf{x}_1 - \mathbf{x}^\star\|^2 - \sum_{t=1}^{T} \frac{(1 - C_1)\lambda A_t}{2} \|\mathbf{x}_t - \mathbf{x}_{t+1}\|^2$$

---

[3]We note that the choice of $M_1$ relies on $\mathbf{x}_1$ because $\mathbf{x}_1$ is the initial point of the algorithm, which is known.

As for the other term of $\sum_{t=1}^{T} \alpha_t \langle \nabla \widehat{f}(\bar{\mathbf{x}}_t) - \nabla \widehat{f}(\widetilde{\mathbf{x}}_t), \mathbf{x}_t - \mathbf{x}_{t-1} \rangle$ in Theorem 2, we have

$$\sum_{t=1}^{T} \alpha_t \langle \nabla \widehat{f}(\bar{\mathbf{x}}_t) - \nabla \widehat{f}(\widetilde{\mathbf{x}}_t), \mathbf{x}_t - \mathbf{x}_{t-1} \rangle \leq \sum_{t=1}^{T} \alpha_t \left\| \nabla \widehat{f}(\bar{\mathbf{x}}_t) - \nabla \widehat{f}(\widetilde{\mathbf{x}}_t) \right\| \left\| \mathbf{x}_t - \mathbf{x}_{t-1} \right\|$$

$$= \sum_{t=1}^{T} \alpha_t \left\| \nabla f(\bar{\mathbf{x}}_t) - \nabla f(\widetilde{\mathbf{x}}_t) - \lambda \bar{\mathbf{x}}_t + \lambda \widetilde{\mathbf{x}}_t \right\| \left\| \mathbf{x}_t - \mathbf{x}_{t-1} \right\|$$

$$\leq \sum_{t=1}^{T} \alpha_t \left( \left\| \nabla f(\bar{\mathbf{x}}_t) - \nabla f(\widetilde{\mathbf{x}}_t) \right\| + \lambda \left\| \bar{\mathbf{x}}_t - \widetilde{\mathbf{x}}_t \right\| \right) \left\| \mathbf{x}_t - \mathbf{x}_{t-1} \right\|$$

$$\leq 2L \sum_{t=1}^{T} \alpha_t \left\| \bar{\mathbf{x}}_t - \widetilde{\mathbf{x}}_t \right\| \left\| \mathbf{x}_t - \mathbf{x}_{t-1} \right\| = 2L \sum_{t=1}^{T} \frac{\alpha_t^2}{A_t} \left\| \mathbf{x}_t - \mathbf{x}_{t-1} \right\|^2$$

where the second step is because of the definition of $\widehat{f}(\cdot)$: $\nabla \widehat{f}(\mathbf{x}) = \nabla f(\mathbf{x}) - \lambda \mathbf{x}$ for any $\mathbf{x} \in \mathbb{R}^d$, the fourth step is due to smoothness (Assumption 2) and the fact of $\lambda \leq L$, and the last step leverages the useful equation of Eq. (15). Combining all terms, we obtain

$$A_T \left[ f(\bar{\mathbf{x}}_T) - f(\mathbf{x}^{\star}) \right] \leq \sum_{t=1}^{T} \left( \frac{2L\alpha_t^2}{A_t} - \frac{(1-C_1)\lambda A_{t-1}}{2} \right) \left\| \mathbf{x}_t - \mathbf{x}_{t-1} \right\|^2 + \frac{\lambda \alpha_1}{2} \left\| \mathbf{x}_1 - \mathbf{x}^{\star} \right\|^2$$

$$+ \sum_{t=2}^{T} \left( \frac{\lambda \alpha_t^2}{2C_0 A_t} + \frac{\lambda A_{t-1}(2C_0 - C_1)}{2} \right) \left\| \mathbf{x}_t - \mathbf{x}_{t-1} \right\|^2.$$

To cancel the positive terms in terms of $\|\mathbf{x}_t - \mathbf{x}_{t-1}\|^2$, their coefficients should satisfy the condition:

$$\frac{2L\alpha_t^2}{A_t} - \frac{(1-C_1)\lambda A_{t-1}}{2} \leq 0, \quad \frac{\lambda \alpha_t^2}{2C_0 A_t} + \frac{\lambda A_{t-1}(2C_0 - C_1)}{2} \leq 0,$$

which is equivalent to

$$\frac{\alpha_t^2}{A_t A_{t-1}} \leq \min \left\{ \frac{1-C_1}{4\kappa}, (C_1 - 2C_0)C_0 \right\}. \tag{19}$$

By setting $C_0 = \frac{C_1}{4}$ and $C_1 = \frac{-1+\sqrt{1+2\kappa}}{\kappa}$, and because

$$\min \left\{ \frac{1-C_1}{4\kappa}, (C_1 - 2C_0)C_0 \right\} = \frac{(-1+\sqrt{1+2\kappa})^2}{8\kappa^2},$$

by setting $\alpha_t = CA_{t-1}$, the following inequality is a sufficient condition for (19):

$$\frac{\alpha_t^2}{A_{t-1}A_t} = \frac{C^2}{C+1} = \frac{(-1+\sqrt{1+2\kappa})^2}{8\kappa^2} \Rightarrow 8\kappa^2 C^2 - XC - X = 0, \text{ where } X = (-1+\sqrt{1+2\kappa})^2.$$

Solving the quadratic equation, we have

$$C = \frac{X + \sqrt{X^2 + 32\kappa^2 X}}{16\kappa^2}.$$

Thus we have $A_t = A_{t-1} + \alpha_t = (C+1)A_{t-1} = (C+1)^{t-1}\alpha_1$, i.e., $\alpha_1/A_T = (C+1)^{-(T-1)}$. More specifically, for any $t \in [T]$, we have

$$\left( \frac{1}{C+1} \right)^t = \left( \frac{1}{1 + \frac{X+\sqrt{X^2+32\kappa^2 X}}{16\kappa^2}} \right)^t = \left( \frac{16\kappa^2}{16\kappa^2 + X + \sqrt{X^2+32\kappa^2 X}} \right)^t$$

$$= \left( 1 - \frac{X + \sqrt{X^2+32\kappa^2 X}}{16\kappa^2 + X + \sqrt{X^2+32\kappa^2 X}} \right)^t \leq \exp \left( \frac{-t}{1 + \frac{16\kappa^2}{X+\sqrt{X^2+32\kappa^2 X}}} \right)$$

$$\leq \exp \left( -\frac{t}{1 + \frac{16\kappa^2}{\sqrt{32\kappa^2 X}}} \right) = \exp \left( -\frac{t}{1 + \frac{4\kappa}{\sqrt{2X}}} \right)$$

$$= \exp \left( -\frac{t}{1 + \frac{4\kappa}{\sqrt{2(-1+\sqrt{1+2\kappa})^2}}} \right) \leq \exp \left( -\frac{t}{1 + \frac{4\kappa}{\sqrt{2(1+2\kappa)}}} \right) \leq \exp \left( -\frac{t}{1 + 2\sqrt{\kappa}} \right).$$

which completes the proof. □

### B.5 Proof of Corollary 2

Before providing the proof, we list two useful lemmas for the AdaGrad-type step size.

**Lemma 3** (Lemma 3.5 of Auer et al. [2002]). *Let $a_1, a_2, \ldots, a_T$ and $\delta$ be non-negative real numbers. Then, it holds that*

$$\sum_{t=1}^{T} \frac{a_t}{\sqrt{\delta + \sum_{s=1}^{t} a_s}} \leq 2\sqrt{\delta + \sum_{t=1}^{T} a_t}, \quad \text{where } 0/\sqrt{0} = 0.$$

**Lemma 4** (Lemma 27 of Kreisler et al. [2024]). *For any positive number $c_1$ and $c_2$, for any $t \geq 0$, and for any sequence of of non-negative numbers $B_0, B_1, B_2, \ldots$, we have*

$$c_1 \sqrt{\sum_{s=0}^{t} B_s^2} - \sum_{s=0}^{t} \frac{B_s^2}{c_2} \sqrt{\sum_{k=0}^{s} B_k^2} \leq 2c_1^{3/2} c_2^{1/2}.$$

*Proof of Corollary 2.* For completeness, we restate the main results of Theorem 1 as follows:

$$A_T \left[ f(\bar{\mathbf{x}}_T) - f(\mathbf{x}^\star) \right] \leq \underbrace{\sum_{t=1}^{T} \langle \alpha_t \nabla f(\widetilde{\mathbf{x}}_t), \mathbf{x}_t - \mathbf{x}^\star \rangle}_{\text{TERM (A)}} + \underbrace{\sum_{t=1}^{T} \alpha_t \langle \nabla f(\widetilde{\mathbf{x}}_t) - \nabla f(\bar{\mathbf{x}}_t), \mathbf{x}_{t-1} - \mathbf{x}_t \rangle}_{\text{TERM (B)}}.$$

For TERM (B), we decompose it into two parts:

$$\text{TERM (B)} \leq \sum_{t=1}^{T} \eta_t \alpha_t^2 \|\nabla f(\widetilde{\mathbf{x}}_t) - \nabla f(\bar{\mathbf{x}}_t)\|^2 + \sum_{t=1}^{T} \frac{1}{4\eta_t} \|\mathbf{x}_t - \mathbf{x}_{t-1}\|^2,$$

using AM-GM inequality: $\sqrt{xy} \leq \frac{x}{2a} + \frac{ay}{2}$ for any $x, y, a > 0$. For TERM (A), using the Bregman proximal inequality Lemma 1, we have

$$\text{TERM (A)} \leq \sum_{t=1}^{T} \frac{1}{2\eta_{t-1}} \left( \|\mathbf{x}_{t-1} - \mathbf{x}^\star\|^2 - \|\mathbf{x}_t - \mathbf{x}^\star\|^2 \right) - \sum_{t=1}^{T} \frac{1}{2\eta_{t-1}} \|\mathbf{x}_t - \mathbf{x}_{t-1}\|^2$$

$$\leq \sum_{t=1}^{T-1} \left( \frac{1}{2\eta_t} - \frac{1}{2\eta_{t-1}} \right) \|\mathbf{x}_t - \mathbf{x}^\star\|^2 - \sum_{t=1}^{T} \frac{1}{2\eta_t} \|\mathbf{x}_t - \mathbf{x}_{t-1}\|^2 + \sum_{t=1}^{T} \left( \frac{1}{2\eta_t} - \frac{1}{2\eta_{t-1}} \right) \|\mathbf{x}_t - \mathbf{x}_{t-1}\|^2$$

$$\leq \frac{D^2}{\eta_T} - \sum_{t=1}^{T} \frac{1}{2\eta_t} \|\mathbf{x}_t - \mathbf{x}_{t-1}\|^2,$$

where the last step uses the boundedness of the domain (Assumption 1). Combining both terms,

$$A_T \left[ f(\bar{\mathbf{x}}_T) - f(\mathbf{x}^\star) \right] \leq \frac{D^2}{\eta_T} + \sum_{t=1}^{T} \eta_t \alpha_t^2 \|\nabla f(\widetilde{\mathbf{x}}_t) - \nabla f(\bar{\mathbf{x}}_t)\|^2 - \sum_{t=1}^{T} \frac{1}{4\eta_t} \|\mathbf{x}_t - \mathbf{x}_{t-1}\|^2. \quad (20)$$

In the following, we consider smooth and non-smooth cases separately.

**Smoothness Case.** Using smoothness (Assumption 2) and the setup of $\alpha_t = t$, we have

$$\|\nabla f(\widetilde{\mathbf{x}}_t) - \nabla f(\bar{\mathbf{x}}_t)\|^2 \leq L^2 \|\widetilde{\mathbf{x}}_t - \bar{\mathbf{x}}_t\|^2 = \frac{L^2 \alpha_t^2}{A_t^2} \|\mathbf{x}_t - \mathbf{x}_{t-1}\|^2 = \frac{4L^2 t^2}{t^2 (t+1)^2} \|\mathbf{x}_t - \mathbf{x}_{t-1}\|^2$$

$$= \frac{4L^2}{\alpha_{t+1}^2} \|\mathbf{x}_t - \mathbf{x}_{t-1}\|^2 \leq \frac{4L^2}{\alpha_t^2} \|\mathbf{x}_t - \mathbf{x}_{t-1}\|^2,$$

where the second step uses the useful equation of Eq. (15). Consequently, we obtain

$$A_T \left[ f(\bar{\mathbf{x}}_T) - f(\mathbf{x}^\star) \right] \leq \frac{D^2}{\eta_T} + \sum_{t=1}^{T} \left( \eta_t - \frac{1}{16L^2 \eta_t} \right) \alpha_t^2 \|\nabla f(\widetilde{\mathbf{x}}_t) - \nabla f(\bar{\mathbf{x}}_t)\|^2$$

$$\leq 3D\sqrt{\sum_{t=1}^{T}\alpha_t^2\|\nabla f(\widetilde{\mathbf{x}}_t) - \nabla f(\bar{\mathbf{x}}_t)\|^2} - \sum_{t=1}^{T}\frac{\alpha_t^2\|\nabla f(\widetilde{\mathbf{x}}_t) - \nabla f(\bar{\mathbf{x}}_t)\|^2}{16L^2D}\sqrt{\sum_{s=1}^{t}\alpha_s^2\|\nabla f(\widetilde{\mathbf{x}}_s) - \nabla f(\bar{\mathbf{x}}_s)\|^2}$$

$$\leq \mathcal{O}\left(LD^2\right),$$

where the second step uses Lemma 3 and the last step uses Lemma 4 with $c_1 = 3D$ and $c_2 = 16L^2D$, resulting in the final bound of $\mathcal{O}\left(LD^2/T^2\right)$.

**Non-Smoothness Case.** Starting from (20), we have

$$A_T\left[f(\bar{\mathbf{x}}_T) - f(\mathbf{x}^\star)\right] \leq \frac{D^2}{\eta_T} + \sum_{t=1}^{T}\eta_t\alpha_t^2\|\nabla f(\widetilde{\mathbf{x}}_t) - \nabla f(\bar{\mathbf{x}}_t)\|^2 \leq 3D\sqrt{\sum_{t=1}^{T}\alpha_t^2\|\nabla f(\widetilde{\mathbf{x}}_t) - \nabla f(\bar{\mathbf{x}}_t)\|^2}$$

$$\leq 3D\sqrt{2G^2\sum_{t=1}^{T}\alpha_t^2} = \mathcal{O}\left(GDT^{3/2}\right).$$

where the second step uses Lemma 3, the third step uses the assumption of bounded gradients: $\|\nabla f(\cdot)\| \leq G$, and the last step is due to the fact of $\sum_{t=1}^{T}\alpha_t^2 = \sum_{t=1}^{T}t^2 = \mathcal{O}(T^3)$. Diving both sides by $A_T = \Theta(T^2)$, we obtain the final bound of $\mathcal{O}(GD/\sqrt{T})$, completing the proof. $\qquad\square$

### B.6  Proof of Theorem 4

*Proof.* To start with, we analyze the intermediate term of:

$$\sum_{t=1}^{T}\alpha_t\left[f(\widetilde{\mathbf{x}}_t) - f(\mathbf{x}^\star)\right] \leq \sum_{t=1}^{T}\langle\alpha_t\nabla f(\widetilde{\mathbf{x}}_t), \mathbf{x}_{t-1} - \mathbf{x}^\star\rangle + \sum_{t=2}^{T}\langle\alpha_t\nabla f(\widetilde{\mathbf{x}}_t), \widetilde{\mathbf{x}}_t - \mathbf{x}_{t-1}\rangle, \qquad (21)$$

where we omit the index of $t = 1$ because $\widetilde{\mathbf{x}}_1 = \mathbf{x}_0$. Via the useful equation of $A_{t-1}\left(\widetilde{\mathbf{x}}_t - \widetilde{\mathbf{x}}_{t-1}\right) = \alpha_t\left(\mathbf{x}_{t-1} - \widetilde{\mathbf{x}}_t\right) + \alpha_{t-1}\left(\mathbf{x}_{t-1} - \mathbf{x}_{t-2}\right)$ (a detailed derivation is deferred to Appendix A due to page limits), the second term above can be bounded as

$$\sum_{t=2}^{T}\langle\alpha_t\nabla f(\widetilde{\mathbf{x}}_t), \widetilde{\mathbf{x}}_t - \mathbf{x}_{t-1}\rangle = \sum_{t=2}^{T}A_{t-1}\langle\nabla f(\widetilde{\mathbf{x}}_t), \widetilde{\mathbf{x}}_{t-1} - \widetilde{\mathbf{x}}_t\rangle + \sum_{t=2}^{T}\langle\alpha_{t-1}\nabla f(\widetilde{\mathbf{x}}_t), \mathbf{x}_{t-1} - \mathbf{x}_{t-2}\rangle$$

$$\leq \sum_{t=2}^{T}A_{t-1}\left[f(\widetilde{\mathbf{x}}_{t-1}) - f(\widetilde{\mathbf{x}}_t)\right] + \sum_{t=2}^{T}\langle\alpha_{t-1}\nabla f(\widetilde{\mathbf{x}}_t), \mathbf{x}_{t-1} - \mathbf{x}_{t-2}\rangle - \sum_{t=2}^{T}A_{t-1}\mathcal{D}_f(\widetilde{\mathbf{x}}_{t-1}, \widetilde{\mathbf{x}}_t),$$

where the last step uses the definition of Bregman divergence. Plugging the above back into (21),

$$A_1f(\widetilde{\mathbf{x}}_1) - A_Tf(\mathbf{x}^\star) \leq \sum_{t=1}^{T}\langle\alpha_t\nabla f(\widetilde{\mathbf{x}}_t), \mathbf{x}_{t-1} - \mathbf{x}^\star\rangle + \sum_{t=2}^{T}\left[A_{t-1}f(\widetilde{\mathbf{x}}_{t-1}) - A_tf(\widetilde{\mathbf{x}}_t)\right]$$

$$+ \sum_{t=2}^{T}\langle\alpha_{t-1}\nabla f(\widetilde{\mathbf{x}}_t), \mathbf{x}_{t-1} - \mathbf{x}_{t-2}\rangle - \sum_{t=2}^{T}A_{t-1}\mathcal{D}_f(\widetilde{\mathbf{x}}_{t-1}, \widetilde{\mathbf{x}}_t).$$

As a result, we can upper-bound $A_T\left[f(\widetilde{\mathbf{x}}_T) - f(\mathbf{x}^\star)\right]$ by

$$\sum_{t=1}^{T}\langle\alpha_t\nabla f(\widetilde{\mathbf{x}}_t), \mathbf{x}_{t-1} - \mathbf{x}^\star\rangle + \sum_{t=2}^{T}\langle\alpha_{t-1}\nabla f(\widetilde{\mathbf{x}}_t), \mathbf{x}_{t-1} - \mathbf{x}_{t-2}\rangle - \sum_{t=2}^{T}A_{t-1}\mathcal{D}_f(\widetilde{\mathbf{x}}_{t-1}, \widetilde{\mathbf{x}}_t).$$

Compared with the final result in Theorem 4, we need to handle the differential terms of

$$\sum_{t=1}^{T}\langle\alpha_t\nabla f(\widetilde{\mathbf{x}}_t), \mathbf{x}_{t-1} - \mathbf{x}_t\rangle + \sum_{t=2}^{T}\langle\alpha_{t-1}\nabla f(\widetilde{\mathbf{x}}_t), \mathbf{x}_{t-1} - \mathbf{x}_{t-2}\rangle = \sum_{t=1}^{T}\langle\alpha_t\nabla f(\widetilde{\mathbf{x}}_t), \mathbf{x}_{t-1} - \mathbf{x}_t\rangle$$

$$+ \sum_{t=1}^{T-1}\langle\alpha_t\nabla f(\widetilde{\mathbf{x}}_{t+1}), \mathbf{x}_t - \mathbf{x}_{t-1}\rangle = \sum_{t=1}^{T-1}\alpha_t\langle\nabla f(\widetilde{\mathbf{x}}_{t+1}) - \nabla f(\widetilde{\mathbf{x}}_t), \mathbf{x}_t - \mathbf{x}_{t-1}\rangle,$$

where the second step shifts the time index, and the last step is because of $\alpha_T = 0$. $\qquad\square$

## B.7 Proof of Theorem 5

Before providing the proof, we present a useful property of smoothness.

**Proposition 2** (Theorem 2.1.5 of Nesterov [2018]). $f(\cdot)$ is $L$-smooth over $\mathbb{R}^d$ if and only if

$$\|\nabla f(\mathbf{x}) - \nabla f(\mathbf{y})\|^2 \leq 2L \cdot \mathcal{D}_f(\mathbf{y}, \mathbf{x}), \quad \text{for any } \mathbf{x}, \mathbf{y} \in \mathbb{R}^d. \tag{22}$$

*Proof of Theorem 5.* For completeness, we restate the main results of Theorem 4 as follows:

$$A_T\left[f(\widetilde{\mathbf{x}}_T) - f(\mathbf{x}^\star)\right] \leq \underbrace{\sum_{t=1}^{T}\langle \alpha_t \nabla f(\widetilde{\mathbf{x}}_t), \mathbf{x}_t - \mathbf{x}^\star\rangle}_{\text{TERM (A)}}$$

$$+ \underbrace{\sum_{t=1}^{T-1}\alpha_t\langle \nabla f(\widetilde{\mathbf{x}}_{t+1}) - \nabla f(\widetilde{\mathbf{x}}_t), \mathbf{x}_t - \mathbf{x}_{t-1}\rangle}_{\text{TERM (B)}} - \sum_{t=2}^{T}A_{t-1}\mathcal{D}_f(\widetilde{\mathbf{x}}_{t-1}, \widetilde{\mathbf{x}}_t).$$

For TERM (B), it holds that

$$\text{TERM (B)} \leq \frac{1}{2}\sum_{t=1}^{T-1}\eta_t\alpha_t^2\|\nabla f(\widetilde{\mathbf{x}}_{t+1}) - \nabla f(\widetilde{\mathbf{x}}_t)\|^2 + \sum_{t=1}^{T-1}\frac{1}{2\eta_t}\|\mathbf{x}_t - \mathbf{x}_{t-1}\|^2$$

$$\leq D\sqrt{\sum_{t=1}^{T-1}\alpha_t^2\|\nabla f(\widetilde{\mathbf{x}}_{t+1}) - \nabla f(\widetilde{\mathbf{x}}_t)\|^2} + \sum_{t=1}^{T-1}\frac{1}{2\eta_t}\|\mathbf{x}_t - \mathbf{x}_{t-1}\|^2,$$

where the first step uses AM-GM inequality and the second step holds due to the step sizes (11) and the self-confident tuning lemma (Lemma 3). As for TERM (A), following the same analysis as in Appendix B.5, using Lemma 1, it holds that

$$\text{TERM (A)} \leq \frac{D^2}{\eta_{T-1}} - \sum_{t=1}^{T-1}\frac{1}{2\eta_t}\|\mathbf{x}_t - \mathbf{x}_{t-1}\|^2.$$

Combining the above results, it holds that

$$A_T\left[f(\widetilde{\mathbf{x}}_T) - f(\mathbf{x}^\star)\right] \leq 2D\sqrt{\sum_{t=1}^{T-1}\alpha_t^2\|\nabla f(\widetilde{\mathbf{x}}_{t+1}) - \nabla f(\widetilde{\mathbf{x}}_t)\|^2} - \sum_{t=2}^{T}A_{t-1}\mathcal{D}_f(\widetilde{\mathbf{x}}_{t-1}, \widetilde{\mathbf{x}}_t). \tag{23}$$

In the following, we consider smooth and non-smooth cases separately.

**Smoothness Case.** Starting from (23), we obtain

$$A_T\left[f(\widetilde{\mathbf{x}}_T) - f(\mathbf{x}^\star)\right] \leq 2D\sqrt{2L\sum_{t=1}^{T-1}\alpha_t^2\mathcal{D}_f(\widetilde{\mathbf{x}}_t, \widetilde{\mathbf{x}}_{t+1})} - \sum_{t=1}^{T-1}A_t\mathcal{D}_f(\widetilde{\mathbf{x}}_t, \widetilde{\mathbf{x}}_{t+1})$$

$$\leq 2\gamma LD + \sum_{t=1}^{T-1}\left(\frac{\alpha_t^2 D}{\gamma} - A_t\right)\mathcal{D}_f(\widetilde{\mathbf{x}}_t, \widetilde{\mathbf{x}}_{t+1}) \leq \mathcal{O}(LD^2),$$

where the first step uses the property of smoothness in Proposition 2, the second step uses AM-GM inequality $\sqrt{xy} \leq \frac{x}{2\gamma} + \frac{\gamma y}{2}$ for any $x, y, \gamma > 0$, and the last step holds by simply choosing $\gamma = 2D$ (this constant only appears in the analysis and thus can be choosen arbitrarily). Finally, since $A_T = \frac{T(T-1)}{2} = \Theta(T^2)$, we achieve the optimal accelerated rate of $\mathcal{O}(LD^2/T^2)$.

**Non-Smoothness Case.** Starting from (23), we have

$$A_T\left[f(\widetilde{\mathbf{x}}_T) - f(\mathbf{x}^\star)\right] \leq 2D\sqrt{\sum_{t=1}^{T-1}\alpha_t^2\|\nabla f(\widetilde{\mathbf{x}}_{t+1}) - \nabla f(\widetilde{\mathbf{x}}_t)\|^2} \leq 4D\sqrt{G^2\sum_{t=1}^{T}\alpha_t^2} = \mathcal{O}\left(GDT^{3/2}\right),$$

the second step uses the assumption of bounded gradients: $\|\nabla f(\cdot)\| \leq G$, and the last step is due to the fact of $\sum_{t=1}^{T}\alpha_t^2 = \sum_{t=1}^{T}t^2 = \mathcal{O}(T^3)$. Diving both sides by $A_T = \Theta(T^2)$, we obtain the final bound of $\mathcal{O}(GD/\sqrt{T})$, completing the proof. □

# C  Extension to Stochastic Optimization

In this section, we extend our proposed methods to the stochastic optimization setting. In the following, we first provide a formal description of the stochastic gradient oracle.

**Assumption 3** (Stochastic Gradient Oracle). For the objective function $f : \mathcal{X} \to \mathbb{R}$, we assume access to a stochastic gradient oracle $g(\cdot)$ that satisfies:

*(i)* Unbiasedness: $\mathbb{E}[\mathbf{g}(\mathbf{x}) \mid \mathbf{x}] = \nabla f(\mathbf{x})$ for all $\mathbf{x} \in \mathcal{X}$;

*(ii)* Bounded noise: $\|\mathbf{g}(\mathbf{x}) - \nabla f(\mathbf{x})\| \leq \sigma$ almost surely, where $\sigma > 0$ is the noise parameter.

Note that Assumption 3 leverages an almost-sure boundedness because we aim to achieve *high-probability* rates, which is stronger than the expectation rates obtained by previous works [Cutkosky, 2019, Kavis et al., 2019, Joulani et al., 2020b], where a bounded-variance assumption suffices.

Below we present our result for the stochastic optimization setting.

**Corollary 3.** *Under Assumptions 1 and 3, our Algorithm 2 with weights $\alpha_t = t$ for $t \in [T-1]$, $\alpha_T = 0$, and with step size*

$$\eta_t = \frac{D}{\sqrt{\sum_{s=1}^{t} \alpha_s^2 \|\mathbf{g}(\widetilde{\mathbf{x}}_{s+1}) - \mathbf{g}(\widetilde{\mathbf{x}}_s)\|^2}}, \tag{24}$$

*with probability at least $1 - \delta$, guarantees*

$$f(\widetilde{\mathbf{x}}_T) - f(\mathbf{x}^*) \leq \begin{cases} \mathcal{O}\left( \frac{LD^2}{T^2} + \frac{\sigma D \sqrt{\theta_{T,\delta/3}}}{\sqrt{T}} + \frac{\sigma D \theta_{T,\delta/3}}{T} \right), & \text{if } f(\cdot) \text{ satisfies Assumption 2,} \\ \mathcal{O}\left( \frac{D(G+\sigma)}{\sqrt{T}} + \frac{\sigma D \sqrt{\theta_{T,\delta/3}}}{\sqrt{T}} + \frac{\sigma D \theta_{T,\delta/3}}{T} \right), & \text{if } \|\nabla f(\cdot)\| \leq G, \end{cases}$$

*where $\theta_{t,\delta/3} = \log \frac{180 \log(6t)}{\delta}$ for any $t \in [T]$ and $\delta > 0$.*

Before presenting the proof of Corollary 3, we import a useful lemma from Ivgi et al. [2023].

**Lemma 5** (Lemma D.2 of Ivgi et al. [2023]). *Let $c > 0$, $\{X_t\}$ be a martingale difference sequence adapted to filtration $\mathcal{F}_t$ with $\mathbb{E}[X_t] = 0$ and $|X_t| \leq c$, and $\{y_t\}$ be a non-negative and non-decreasing sequence. Then, for any $\delta \in (0,1)$, with probability at least $1 - \delta$, we have*

$$\left| \sum_{t=1}^{T} y_t X_t \right| \leq 8 y_T \sqrt{\theta_{T,\delta} \sum_{t=1}^{T} X_t^2 + c^2 \theta_{T,\delta}^2},$$

*where $\theta_{t,\delta} \triangleq \log \frac{60 \log(6t)}{\delta}$.*

*Proof of Corollary 3.* We start from the beginning of Theorem 5 and aim to analyze two main terms:

$$\text{TERM (A)} = \sum_{t=1}^{T} \langle \alpha_t \nabla f(\widetilde{\mathbf{x}}_t), \mathbf{x}_t - \mathbf{x}^* \rangle,$$

$$\text{and TERM (B)} = \sum_{t=1}^{T-1} \alpha_t \langle \nabla f(\widetilde{\mathbf{x}}_{t+1}) - \nabla f(\widetilde{\mathbf{x}}_t), \mathbf{x}_t - \mathbf{x}_{t-1} \rangle.$$

For TERM (A), we further decompose it into two parts because we can only access the stochastic gradient $\mathbf{g}(\widetilde{\mathbf{x}}_t)$:

$$\text{TERM (A)} = \underbrace{\sum_{t=1}^{T} \langle \alpha_t \mathbf{g}(\widetilde{\mathbf{x}}_t), \mathbf{x}_t - \mathbf{x}^* \rangle}_{\text{TERM (A-I)}} + \underbrace{\sum_{t=1}^{T-1} \alpha_t \langle \nabla f(\widetilde{\mathbf{x}}_t) - \mathbf{g}(\widetilde{\mathbf{x}}_t), \mathbf{x}_t - \mathbf{x}^* \rangle}_{\text{TERM (A-II)}},$$

where TERM (A-II) only counts from $t = 1$ to $t = T - 1$ because we manually set $\alpha_T = 0$. For TERM (A-I), we can use the Bregman proximal inequality (Lemma 1) to obtain

$$\text{TERM (A-I)} \leq \frac{D^2}{\eta_{T-1}} - \sum_{t=1}^{T-1} \frac{1}{2\eta_t} \|\mathbf{x}_t - \mathbf{x}_{t-1}\|^2.$$

And for TERM (A-II), using the martingale concentration inequality (Lemma 5) with

$$X_t = \left\langle \nabla f(\widetilde{\mathbf{x}}_t) - \mathbf{g}(\widetilde{\mathbf{x}}_t), \frac{\mathbf{x}_t - \mathbf{x}^\star}{D} \right\rangle, \quad y_t = \alpha_t D, \quad c = \sigma,$$

where $(\mathcal{F}_t)_{t\geq 0}$ denotes the filtration generated by the history up to round $t$, i.e., $\mathcal{F}_t = \sigma\left(\mathbf{x}_0, \{\mathbf{x}_s, \widetilde{\mathbf{x}}_s, \mathbf{g}(\widetilde{\mathbf{x}}_s)\}_{s=1}^t\right)$, and $\{X_t\}_{t=1}^{T-1}$ is a martingale difference sequence adapted to $(\mathcal{F}_t)_{t\geq 0}$; with probability at least $1 - \delta/3$,

$$\text{TERM (A-II)} \leq 8(T-1)D\sqrt{\theta_{T,\delta/3}\sigma^2(T-1) + \sigma^2\theta_{T,\delta/3}^2},$$

where $\theta_{t,\delta/3} = \log\frac{180\log(6t)}{\delta}$. For TERM (B), we can decompose it into two parts of

$$\text{TERM (B-I)} = \sum_{t=1}^{T-1} \alpha_t \langle \mathbf{g}(\widetilde{\mathbf{x}}_{t+1}) - \mathbf{g}(\widetilde{\mathbf{x}}_t), \mathbf{x}_t - \mathbf{x}_{t-1} \rangle,$$

and $\text{TERM (B-II)} = \sum_{t=1}^{T-1} \alpha_t \langle \nabla f(\widetilde{\mathbf{x}}_{t+1}) - \mathbf{g}(\widetilde{\mathbf{x}}_{t+1}), \mathbf{x}_t - \mathbf{x}_{t-1} \rangle + \sum_{t=1}^{T-1} \langle \mathbf{g}(\widetilde{\mathbf{x}}_t) - \nabla f(\widetilde{\mathbf{x}}_t), \mathbf{x}_t - \mathbf{x}_{t-1} \rangle.$

For TERM (B-I), following the same analysis as Theorem 5, we have

$$\text{TERM (B-I)} \leq D\sqrt{\sum_{t=1}^{T-1} \alpha_t^2 \|\mathbf{g}(\widetilde{\mathbf{x}}_{t+1}) - \mathbf{g}(\widetilde{\mathbf{x}}_t)\|^2} + \sum_{t=1}^{T-1} \frac{1}{2\eta_t}\|\mathbf{x}_t - \mathbf{x}_{t-1}\|^2.$$

And for TERM (B-II), we can use the martingale concentration inequality (Lemma 5) again, where we note that $X_t = \langle \nabla f(\widetilde{\mathbf{x}}_{t+1}) - \mathbf{g}(\widetilde{\mathbf{x}}_{t+1}), \mathbf{x}_t - \mathbf{x}_{t-1}\rangle/D$ is also a martingale difference sequence adapted to the filtration $(\mathcal{F}_t)_{t\geq 0}$ because $\widetilde{\mathbf{x}}_{t+1}$ only contains the information up to the $t$-th iteration. Thus, with probability at least $1 - 2\delta/3$, we have

$$\text{TERM (B-II)} \leq 16(T-1)D\sqrt{\theta_{T-1,\delta/3}\sigma^2(T-1) + \sigma^2\theta_{T-1,\delta/3}^2}.$$

In the following, we consider smooth and non-smooth cases separately.

**Smoothness Case.** Combining TERM (A-I), TERM (B-I), and the negative Bregman divergence as shown in Theorem 4, we have

$$2D\sqrt{\sum_{t=1}^{T-1} \alpha_t^2 \|\mathbf{g}(\widetilde{\mathbf{x}}_{t+1}) - \mathbf{g}(\widetilde{\mathbf{x}}_t)\|^2} - \sum_{t=1}^{T-1} A_t \mathcal{D}_f(\widetilde{\mathbf{x}}_t, \widetilde{\mathbf{x}}_{t+1})$$

$$\leq 2D\sqrt{3\sum_{t=1}^{T-1} \alpha_t^2 \|\nabla f(\widetilde{\mathbf{x}}_{t+1}) - \nabla f(\widetilde{\mathbf{x}}_t)\|^2 + 6\sum_{t=1}^{T-1} \alpha_t^2 \sigma^2} - \sum_{t=1}^{T-1} A_t \mathcal{D}_f(\widetilde{\mathbf{x}}_t, \widetilde{\mathbf{x}}_{t+1})$$

$$\lesssim LD^2 + \sigma T^{3/2},$$

because of $\|\nabla f(\mathbf{x}) - \mathbf{g}(\mathbf{x})\| \leq \sigma$ for any $\mathbf{x} \in \mathcal{X}$ and the fact of $\sum_{t=1}^T \alpha_t^2 = \sum_{t=1}^T t^2 = \mathcal{O}(T^3)$. Finally, we can combine all the terms together to obtain the final bound of

$$f(\widetilde{\mathbf{x}}_T) - f(\mathbf{x}^\star) \leq \mathcal{O}\left(\frac{LD^2}{T^2} + \frac{\sigma D\sqrt{\theta_{T,\delta/3}}}{\sqrt{T}} + \frac{\sigma D\theta_{T,\delta/3}}{T}\right),$$

which matches the optimal rate of $\mathcal{O}(LD^2/T^2 + \sigma D/\sqrt{T})$ with only a logarithmic factor overhead. And note that the extra logarithmic factor always appears in high-probability bounds in the literature.

**Non-Smoothness Case.** Combining TERM (A-I) and TERM (B-I), we have

$$2D\sqrt{\sum_{t=1}^{T-1} \alpha_t^2 \|\mathbf{g}(\widetilde{\mathbf{x}}_{t+1}) - \mathbf{g}(\widetilde{\mathbf{x}}_t)\|^2} \leq 2D\sqrt{3\sum_{t=1}^{T-1} \alpha_t^2 \|\nabla f(\widetilde{\mathbf{x}}_{t+1}) - \nabla f(\widetilde{\mathbf{x}}_t)\|^2 + 6\sum_{t=1}^{T-1} \alpha_t^2 \sigma^2}$$

$$\leq 2D\sqrt{6G^2\sum_{t=1}^{T-1}\alpha_t^2 + 6\sigma^2\sum_{t=1}^{T-1}\alpha_t^2} \leq \mathcal{O}\left(D(G+\sigma)T^{3/2}\right),$$

because of $\|\nabla f(\mathbf{x}) - \mathbf{g}(\mathbf{x})\| \leq \sigma$ for any $\mathbf{x} \in \mathcal{X}$ and the fact of $\sum_{t=1}^{T}\alpha_t^2 = \sum_{t=1}^{T}t^2 = \mathcal{O}(T^3)$. Finally, we can combine all the terms together to obtain the final bound of

$$f(\widetilde{\mathbf{x}}_T) - f(\mathbf{x}^\star) \leq \mathcal{O}\left(\frac{D(G+\sigma)}{\sqrt{T}} + \frac{\sigma D\sqrt{\theta_{T,\delta/3}}}{\sqrt{T}} + \frac{\sigma D\theta_{T,\delta/3}}{T}\right),$$

which matches the optimal rate of $\mathcal{O}(D(G+\sigma)/\sqrt{T})$ with only a logarithmic factor overhead. And note that the extra logarithmic factor always appears in high-probability bounds in the literature. $\square$

## D Proof of Proposition 1

*Proof.* We prove this by induction. To begin with, we aim to show $\mathbf{y}_1 = \bar{\mathbf{x}}_1$. To see this, we provide a fundamental analysis for Algorithm 2:

$$
\begin{aligned}
\bar{\mathbf{x}}_t &= \frac{1}{A_t}\left(A_{t-1}\bar{\mathbf{x}}_{t-1} + \alpha_t\mathbf{x}_t\right) = \frac{1}{A_t}\left[A_{t-1}\bar{\mathbf{x}}_{t-1} + \alpha_t\left(\mathbf{x}_{t-1} - \eta_{t-1}\alpha_t\nabla f(\widetilde{\mathbf{x}}_t)\right)\right] \\
&= \frac{1}{A_t}\left[A_{t-1}\bar{\mathbf{x}}_{t-1} + \alpha_t\left(\frac{A_{t-1}\bar{\mathbf{x}}_{t-1} - A_{t-2}\bar{\mathbf{x}}_{t-2}}{\alpha_{t-1}} - \eta_{t-1}\alpha_t\nabla f(\widetilde{\mathbf{x}}_t)\right)\right] \\
&= \bar{\mathbf{x}}_{t-1}\left(\frac{A_{t-1}}{A_t} + \frac{\alpha_t(\alpha_{t-1} + A_{t-2})}{A_t\alpha_{t-1}}\right) - \bar{\mathbf{x}}_{t-2}\left(\frac{\alpha_t A_{t-2}}{A_t\alpha_{t-1}}\right) - \frac{\eta_{t-1}\alpha_t^2}{A_t}\nabla f(\widetilde{\mathbf{x}}_t) \\
&= \bar{\mathbf{x}}_{t-1} + \frac{\alpha_t A_{t-2}}{A_t\alpha_{t-1}}\left(\bar{\mathbf{x}}_{t-1} - \bar{\mathbf{x}}_{t-2}\right) - \frac{\eta_{t-1}\alpha_t^2}{A_t}\nabla f(\widetilde{\mathbf{x}}_t) \\
&= \bar{\mathbf{x}}_{t-1} - \frac{1}{4L}\nabla f(\widetilde{\mathbf{x}}_t) + \frac{t-2}{t+1}\left(\bar{\mathbf{x}}_{t-1} - \bar{\mathbf{x}}_{t-2}\right), \qquad (25)
\end{aligned}
$$

where the second step uses the update rule, and the last step holds due to $\alpha_t = t$, $\eta_{t-1} = \frac{t+1}{t}\cdot\frac{1}{8L}$. For $t = 1$, $A_{t-2} = 0$, we have $\bar{\mathbf{x}}_1 = \bar{\mathbf{x}}_0 - \frac{1}{4L}\nabla f(\widetilde{\mathbf{x}}_1) = \mathbf{x}_0 - \frac{1}{4L}\nabla f(\mathbf{x}_0)$ due to the initialization of $\widetilde{\mathbf{x}}_1 = \bar{\mathbf{x}}_0 = \mathbf{x}_0$. Furthermore, for **NAG** (2), $\mathbf{y}_1 = \mathbf{z}_0 - \theta\nabla f(\mathbf{z}_0) = \mathbf{x}_0 - \frac{1}{4L}\nabla f(\mathbf{x}_0)$ due to the initialization of $\mathbf{y}_0 = \mathbf{z}_0 = \mathbf{x}_0$. To conclude, we have $\mathbf{y}_1 = \bar{\mathbf{x}}_1$.

Consequently, we assume $\mathbf{y}_{t-1} = \bar{\mathbf{x}}_{t-1}$ for some $t > 1$ and aim to prove $\mathbf{z}_{t-1} = \widetilde{\mathbf{x}}_t$. In **NAG** (2), since $\mathbf{z}_{t-1} = \mathbf{y}_{t-1} + \frac{t-2}{t+1}(\mathbf{y}_{t-1} - \mathbf{y}_{t-2})$, we have $\mathbf{z}_{t-1} = \bar{\mathbf{x}}_{t-1} + \frac{t-2}{t+1}(\bar{\mathbf{x}}_{t-1} - \bar{\mathbf{x}}_{t-2})$ due to the induction hypothesis. To prove $\mathbf{z}_{t-1} = \widetilde{\mathbf{x}}_t$, since both of them are linear combinations of the sequence $\{\mathbf{x}_s\}_{s=1}^{t-1}$, we only need to prove that the coefficients of $\mathbf{x}_s$ for all $s \in [t-1]$ are the same. Specifically, for $\mathbf{x}_{t-1}$, its coefficient in $\mathbf{z}_{t-1}$ is $\frac{\alpha_{t-1}}{A_{t-1}}(1 + \frac{t-2}{t+1}) = \frac{2(2t-1)}{t(t+1)}$ and its coefficient in $\widetilde{\mathbf{x}}_t$ is $\frac{1}{A_t}(\alpha_{t-1} + \alpha_t) = \frac{2(2t-1)}{t(t+1)}$, which are the same. For $\mathbf{x}_s$ with $s \in [t-2]$, its coefficient in $\mathbf{z}_{t-1}$ is $\frac{\alpha_s}{A_{t-1}}(1 + \frac{t-2}{t+1}) - \frac{t-2}{t+1}\frac{\alpha_s}{A_{t-2}} = \frac{2\alpha_s}{t(t+1)}$ and its coefficient in $\widetilde{\mathbf{x}}_t$ is $\frac{\alpha_s}{A_t} = \frac{2\alpha_s}{t(t+1)}$, which are the same. As a result, we have proven $\mathbf{z}_{t-1} = \widetilde{\mathbf{x}}_t$.

Finally, we aim to prove $\mathbf{y}_t = \bar{\mathbf{x}}_t$ to finish the proof of induction. To see this, we combine the update of **NAG** (2) into one step and use $\mathbf{y}_{t-1} = \bar{\mathbf{x}}_{t-1}$ and $\mathbf{z}_{t-1} = \widetilde{\mathbf{x}}_t$:

$$
\begin{aligned}
\mathbf{y}_t = \mathbf{z}_{t-1} - \theta\nabla f(\mathbf{z}_{t-1}) &= \left(\mathbf{y}_{t-1} + \frac{t-2}{t+1}(\mathbf{y}_{t-1} - \mathbf{y}_{t-2})\right) - \theta\nabla f(\mathbf{z}_{t-1}) \\
&= \left(\bar{\mathbf{x}}_{t-1} + \frac{t-2}{t+1}(\bar{\mathbf{x}}_{t-1} - \bar{\mathbf{x}}_{t-2})\right) - \frac{1}{4L}\nabla f(\widetilde{\mathbf{x}}_t),
\end{aligned}
$$

which is equivalent to (25), showing that $\mathbf{y}_t = \bar{\mathbf{x}}_t$.

Finally, since we have proved $\widetilde{\mathbf{x}}_t = \bar{\mathbf{x}}_{t-1} + \frac{t-2}{t+1}(\bar{\mathbf{x}}_{t-1} - \bar{\mathbf{x}}_{t-2})$, denoting by $\beta_{t-1} = \frac{t-2}{t+1}$, the update rule of our algorithm can be rewritten as

$$\bar{\mathbf{x}}_t = \bar{\mathbf{x}}_{t-1} + \beta_{t-1}\left(\bar{\mathbf{x}}_{t-1} - \bar{\mathbf{x}}_{t-2}\right) - \frac{1}{4L}\nabla f\left(\bar{\mathbf{x}}_{t-1} + \beta_{t-1}(\bar{\mathbf{x}}_{t-1} - \bar{\mathbf{x}}_{t-2})\right),$$

which is exactly Nesterov's accelerated gradient method in a one-step update formulation. $\square$

# E  Experiments

In this section, we present the numerical experiments to evaluate the performance of the proposed methods. We compare our methods with baseline algorithms across different problem settings (non-universal and universal) on various datasets.

**Contenders.** In the non-universal setting, we compare our method (Algorithm 2 with step size $\eta = \frac{1}{4L}$) with: *(i)* the classic non-universal methods of **NAG**, as stated in Eq. (2); *(ii)* the standard gradient descent **GD**; and *(iii)* the non-universal variant of UniXGrad [Kavis et al., 2019] (whose step size is also set as $\eta = \frac{1}{4L}$). And in the universal setting, we compare our methods — the two-gradient version with adaptive step size Eq. (9) and the one-gradient improvement with adaptive step size Eq. (11) — with the classic universal methods: *(i)* **UniXGrad** [Kavis et al., 2019] and *(ii)* the method in Joulani et al. [2020b] (abbreviated as **JRGS'20**).

**Setup.** Our experiment setup is mainly inspired by Kavis et al. [2019]. We investigate two kinds of convex smooth optimization problems: the squared loss task and the logistic regression task. For the squared loss task, we take the least squares problem with $L_2$-norm ball constraint for this setting, i.e., $f(\mathbf{x}) \triangleq \frac{1}{2N}\|A\mathbf{x} - \mathbf{b}\|_2^2$, where $\|\mathbf{x}\|_2 < R$, $A \in \mathbb{R}^{N \times d}$ follows a normal distribution of $\mathcal{N}(0, \sigma^2 I)$ and $\mathbf{b} = A\mathbf{x}^\star + \varepsilon$ such that $\varepsilon$ is a random vector $\sim \mathcal{N}(\mathbf{0}, 10^{-3})$. We pick $N = 500$ and $d = 100$. The smoothness parameter $L$ of the squared objective is $\frac{1}{N}\sigma_{\max}(A)^2$, where $\sigma_{\max}(A)$ is the largest singular value of $A$.

For the logistic regression task, the performance is measured by the $\ell_2$-regularized logistic loss $f(\mathbf{x}) \triangleq \frac{1}{N}\sum_{i=1}^{N}\log(1 + \exp(-b_i \cdot \mathbf{a}_i^\top \mathbf{x})) + \mu\|\mathbf{x}\|_2^2$, where $\mathbf{a}_t \in \mathbb{R}^d$ and $b_t \in \{-1, +1\}$ are chosen from a dataset $\{\mathbf{a}_t, b_t\}_{i=1}^N$, $\mu = 0.005$ is the parameter of the regularization term to prevent overfitting. The smoothness parameter $L$ of the logistic objective is $\frac{1}{4N}\lambda_{\max}(\sum_{i=1}^{N}\mathbf{a}_i\mathbf{a}_i^\top)$, where $\lambda_{\max}(\cdot)$ is the largest eigenvalue of the matrix. Here we use five LIBSVM datasets to initialize the logistic loss.

In the non-universal setting, all the methods can use the knowledge of the smoothness parameter $L$, which is prohibited in the universal setting.

**Results.** We report average results of the suboptimality gap, i.e., $f(\cdot) - \min_{\mathbf{x} \in \mathcal{X}} f(\mathbf{x})$, in the *logarithmic* scale, and time complexity with standard deviations of 5 independent runs. Only the randomness of the initial point is preserved. All hyper-parameters are set to be theoretically optimal.

Figure 2 plots the suboptimality gap and time complexity of all methods, in the non-universal setting. Our method (Algorithm 2) achieves a similar convergence behavior as **NAG** and **UniXGrad** while being faster than **GD**. As for the time complexity, **GD** is the fastest one, **UniXGrad** is the slowest because there are two gradient evaluations and projections per iteration, while our method and **NAG** are comparable.

Figure 3 plots the suboptimality gap and time complexity of all methods, in the universal setting. Our method (Algorithm 2) achieves comparable and sometimes better convergence behavior compared with the other state-of-the-art methods. In terms of time complexity, our one-gradient improvement has a similar performance as **JRGS'20**, and is faster than **UniXGrad** and our two-gradient version. Note that the standard deviation of the convergence curves become large as the suboptimality gap becomes small because the logarithm scale is used, making it sensitive to the small variations.

The results in Figure 2 and Figure 3 show the effectiveness of our simple method (Algorithm 2) when compared with the classic non-universal methods (**NAG** and **GD**) and the classic universal methods (UniXGrad and the method in Joulani et al. [2020b]), supporting our theoretical findings.

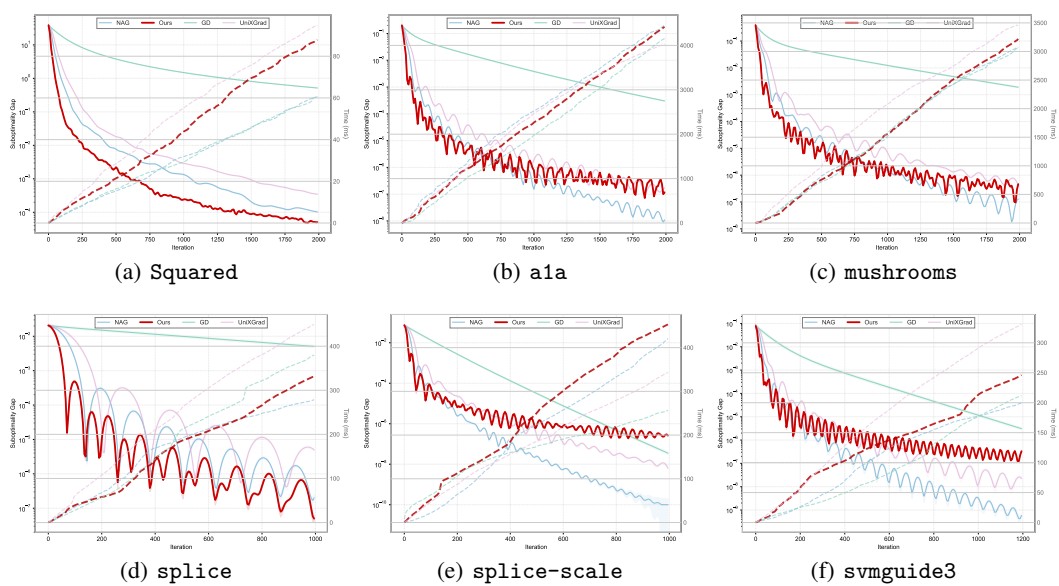

Figure 2: Comparison in the non-universal setting of the convergence curves and time complexity. Our method (**Ours**) is compared with classic non-universal methods **NAG** (2), **GD**, and **UniXGrad** on one squared loss task (Squared) and five $\ell_2$-regularized logistic regression tasks (a1a, mushrooms, splice, splice-scale and svmguide3). **Ours** achieves similar convergence as **NAG** and **UniXGrad** while being faster than **GD**.

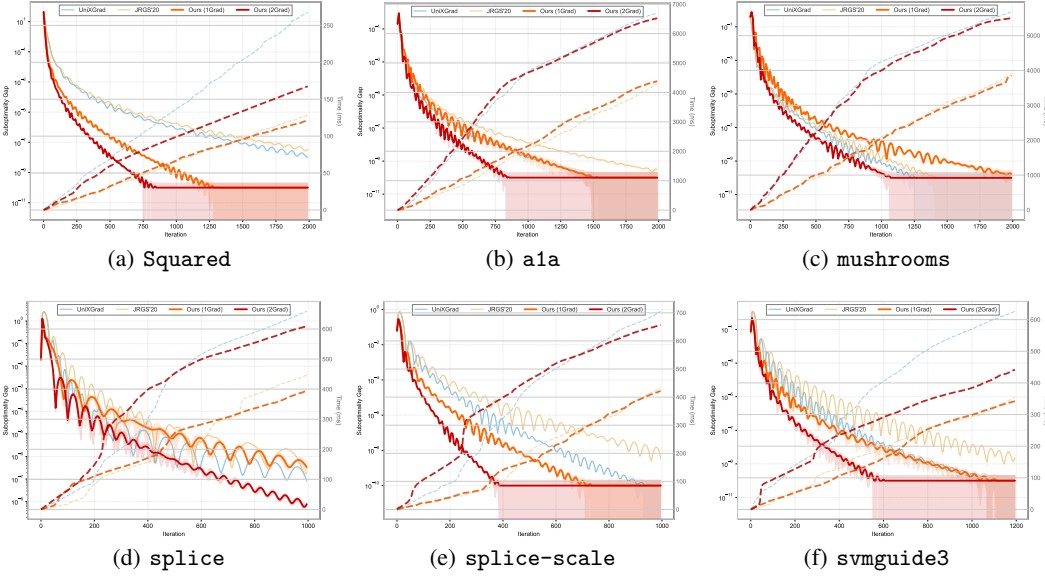

Figure 3: Comparison in the universal setting of the convergence curves and time complexity. Our methods — **Ours (1Grad)** and **Ours (2Grad)** — are compared with classic universal methods **UniXGrad** and **JRGS'20** on one squared loss task (Squared) and five $\ell_2$-regularized logistic regression tasks (a1a, mushrooms, splice, splice-scale and svmguide3). Our methods achieve comparable convergence behavior compared with the other contenders. **Ours (1Grad)** is more efficient than **Ours (2Grad)** and **UniXGrad**.

