# OpenReview forum: "Optimistic Online-to-Batch Conversions for Accelerated Convergence and Universality"
_NeurIPS.cc/2025/Conference — NeurIPS 2025 poster_

### Official Review · Reviewer_h8Yk · 2025-06-18

**Clarity:** 3
**Significance:** 2
**Originality:** 3
**Rating:** 4
**Confidence:** 3

**Summary:**

This paper proposes optimistic online-to-batch conversions, which incorporate the optimistic approach into the online-to-batch stage, enabling the use of non-optimistic methods for achieving the same rate as optimistic methods with stabilized online-to-batch conversions (Cutkosky 2019). This enables the authors to achieve acceleration, both in convex and strongly convex settings, and universality, with a single query per iteration, in contrast to previous work. Additional relations between online approaches, NAG, and the heavy-ball method are discussed.

**Questions:**

1. Can the authors elaborate on what results, if any, were not established by previous work (and cannot be easily established from previous results)?
2. Beyond elegance (and potentially practical use), is the single query per iteration of particular importance beyond saving a constant factor in the convergence rate? I do appreciate the simpler algorithm but missed if there is something beyond simplicity and constant factors.
3. Does the approach enable guarantees in the strongly convex case with stochastic noise? I am aware that the $\sigma / \sqrt{T}$ term cannot be improved. (This is a question out of interest, it is ok if the authors simply do not know the answer.)

Overall, this paper is interesting to read and provides some insights into acceleration and universality, with a somewhat simpler universal algorithm. That being said, articulating new guarantees or benefits achieved from this new perspective would be much appreciated.

**Ethical Concerns:**

["NO or VERY MINOR ethics concerns only"]

**Final Justification:**

As pointed out in the discussion with another reviewer (thanks for the through review!), this paper draws somewhat heavily from,  [WA'18] limiting the significant of this work. That said, the universality result still can be of value to the community, leading me to my limited yet positive support of this paper.

**Quality:**

2

**Strengths And Weaknesses:**

**Strengths**

1. The presentation is well structured, helping the reader understand the difference between existing online-to-batch conversions and the new proposed conversions.
2. The results cover several cases of interest, including convex Lipschitz and convex smooth, with and without stochastic noise, and strongly convex smooth case.

**Weaknesses**

3. My main concern is the novelty of the results. This work mainly offers a new perspective, while each of the results can be obtained from previous work, either directly or with a minor extension (e.g., universal algorithms, Nesterov 2015, Kavis et al. 2019, Rodomanov et al. 2024a,b, or using restarts to support the strongly convex case).

---

> ### Author Rebuttal · Authors · 2025-07-30
>
> We sincerely appreciate your feedback. Below we hope to resolve your concerns about the contributions of our paper.
>
> ---
>
> **W & Q1.** About comparison with previous works and new results.
>
> * This work mainly offers a new perspective, while each of the results can be obtained from previous work, either directly or with a minor extension (e.g., universal algorithms, Nesterov 2015, Kavis et al. 2019, Rodomanov et al. 2024a,b, or using restarts to support the strongly convex case).
> * Can the authors elaborate on what results, if any, were not established by previous work (and cannot be easily established from previous results)?
>
> **A1.** Thank you for the comments and question. Below we first explain the differences with previous works and then reclarify the main contributions of our work.
>
> * Compared with Nesterov (2015), our universal method holds in the stochastic setup (in Thm 7 in the appendix) while theirs does not. And their line-search based methods cannot be directly extended to the stochastic setup.
> * Compared with Kavis et al. (2019) and Rodomanov et al. (2024a,b), although all the mentioned works can achieve the optimal universal rates, **the involved techniques are completely different**. Besides, our **improved optimistic O2B conversion** allows **the simple OGD with one gradient query** per iteration to achieve the optimal universal rates. This **cannot** be done by the  mentioned works and previous ones.
> * Finally, we would like to note that our algorithm does not need the restarting scheme because we do not consider stochastic gradients for strongly convex objectives.
>
> The contribution of this work is not obtaining new or better convergence rates, as all convergence rates in this work are *classical and optimal*, and thus cannot be improved further. Our main contribution lies in making **a step forward in the line of online-to-batch (O2B) conversion**, where **we propose several powerful optimistic O2B conversions (in Thm 1,3,5) such that using simple OGD can achieve the optimal convergence**, which **cannot** be done via previous O2B-based works. We illustrate the effectiveness of our optimistic O2B conversions in the convex smooth optimization, strongly convex smooth optimization, and the universal setup. Besides, our findings also provide an interesting understanding of the classical NAG from the perspective of online learning, as shown in Sec 4.
>
> ---
>
> **Q2.** Beyond elegance (and potentially practical use), is the single query per iteration of particular importance beyond saving a constant factor in the convergence rate? I do appreciate the simpler algorithm but missed if there is something beyond simplicity and constant factors.
>
> **A2.** Thank you for the question. If we fix the total budget of gradient queries to $T$, reducing the queries per iteration from $2$ to $1$ improves the final rate by a constant factor of $4$ for our results. However, the key messages we aim to convey here are:
>
> 1. **Simplicity is a primary pursuit in the algorithm design**. For example, previous studies (e.g., Kavis et al. (2019), Joulani et al. (2020)) need optimistic algorithms for this purpose while this paper only requires the simple OGD.
> 2. We view this improvement not just as a constant-factor gain, but as a **concrete demonstration of the power of our new optimistic O2B conversion**. It highlights that our framework is more efficient at extracting information to achieve acceleration.
> 3. We achieve the universality to smooth via a simple OGD with one gradient query per iteration, which cannot be done by previous O2B-conversion-based methods.
>
> We will add a clarifying statement to the paper to make our key messages clearer to readers.
>
> ---
>
> **Q3.** Does the approach enable guarantees in the strongly convex case with stochastic noise? I am aware that the $\sigma / \sqrt{T}$ term cannot be improved. (This is a question out of interest, it is ok if the authors simply do not know the answer.)
>
> **A3.** Thanks for the insightful question! We have also been studying this problem recently. Formally, for **$\lambda$-strongly convex** and $L$-smooth objectives with **stochastic** gradients, the optimal convergence rate is $O(\exp(-T/\sqrt{\kappa}) + \sigma/[\lambda T])$, where $\kappa = L/\lambda $ denotes the condition number. Note that the dependence of $T$ in the noise term is $\sigma/T$ but not $\sigma/\sqrt{T}$. Classical methods, e.g., [1,2,3], all require a restarting scheme (as the reviewer mentioned in the 'Weakness' part) to achieve this optimal rate.
>
> In the **O2B-conversion-based** line of research, achieving the optimal convergence is still an **open problem** and highly non-trivial. Therefore, extending our method directly to this setup is challenging and beyond the scope of this work. Actually, we have recently solved this problem, which is not published yet. The solution relies on a substantially different online algorithm design (without the restarting scheme) and novel analyses. It is welcome that you can follow our subsequent works for the progress in this problem :)
>
> ---
>
> We have taken your feedback seriously and will revise the paper to make these points about our contribution clearer. If our responses have properly addressed your concerns, we kindly request a reevaluation of our paper's score. Thank you!
>
> **References:**
>
> [1] Optimal Stochastic Approximation Algorithms for Strongly Convex Stochastic Composite Optimization II Shrinking Procedures and Optimal Algorithms, 2013
>
> [2] Optimal Regularized Dual Averaging Methods for Stochastic Optimization, 2012
>
> [3] A Universally Optimal Multistage Accelerated Stochastic Gradient Method, 2019

---

> ### Comment · Reviewer_h8Yk · 2025-08-02
>
> Thank you for your response.
>
> My original concerns were addressed.
>
> That being said, I will follow on the discussion with Reviewer Wpuw before finalizing my assessment.
>
> After reading the reviews and the responses, if I'm understanding correctly, Algorithm 2 and Theorem 1 are essentially partial components and intermediate results of Wang and Abernethy (2018). It also seems like the trick of using $\hat{f}(x)=f(x)-\frac{\lambda}{2}\lVert x \rVert^2$ is also used by Wang and Abernethy (2018), and the proof of Theorem 4 states that it use a similar analysis to Wang and Abernethy (2018).
>
> While the perspective taken by the two works is different, from a technical perspective, it seems like Section 3.3 is the main part not borrowing heavily from Wang and Abernethy (2018).
>
> Note that I still appreciate the formulation through optimistic conversion and the contribution of Section 3.3.

---

> ### Author Response · Authors · 2025-08-03
>
> Thank you for the detailed follow-up responses and for acknowledging the contribution of Section 3.3. We are glad our previous response addressed your original concerns. Below, we would like to further clarify the technical relationship between our work and Wang and Abernethy (2018) [WA'18], particularly regarding Thms 1-4, to provide a clearer picture of our distinct contributions.
>
> **Q.** About the connection with [WA'18].
>
> **A.** In the following, we clarify the connection of intermediate results in Thms 1,3, and the connection of algorithms in Thms 2,4 to [WA'18], separately.
>
> * **About Thms 1 and 3:** We indeed obtain **the same intermediate results** as those by [WA'18], however, **from entirely different techniques of our proposed optimistic online-to-batch (O2B) conversions**. *We will add a remark to clarify the connection to [WA'18] regarding the intermediate results in the revised version.* Furthermore, **our work's primary technical contribution lies in the technical perspective by proposing novel optimistic O2B conversions**, which we believe is a valuable contribution to the research on O2B-based methods.
>
> * **About Thms 2 and 4:** As explained in the response to Reviewer #Wpuw, **Thms 2 and 4 are corollaries of Thms 1 and 3**, where the convergence rates can be obtained via *standard* online learning algorithms with *common* analyses. Therefore, *we will use "corollaries" in the revised version, and clarify that Thms 2 and 4 are only presented to validate the effectiveness of our proposed optimistic O2B conversions, but not one of our claimed contributions.* Below, we clarify the issues about Thm 2 and 4 separately.
>
>   * **About Thm 4** (Corollary 2 in the revised version): Note that $\hat{f}(x)=f(x)-\frac{\lambda}{2}\\|x\\|^2$ is a **standard** trick (e.g., Lemma 3.11 of [1] and Theorem 2.1.12 of [2]) to transform strongly convex functions into convex functions so that the analysis for the convex case can be reused in the strongly convex case. Therefore, we believe this is a standard technique in strongly convex optimization and will cite [1,2] and [WA'18] in the revised version for clarity. Moreover, in the proof of Thm 4 (Line 489), we have given the proper credits to [WA'18] in terms of the proof flow.
>   * **About Thm 2** (Corollary 1 in the revised version): Since our framework reduces the problem to a look-ahead online learning problem with regret of $\sum_{t=1}^T \langle\alpha_t \nabla f(\tilde{x}_t), x_t-x^*\rangle$, optimizing it requires the standard online gradient descent (OGD) as the algorithm and the standard analysis for OGD.
>
>   We note a parallel with [WA'18], where the main focus is on their Fenchel game formulation, rather than the subsequent algorithmic analysis, which the authors do not claim as a contribution. Similarly, in our work, **the algorithm and its standard analysis are presented mainly to demonstrate the effectiveness of our core contribution: the optimistic O2B conversion framework**.
>
> **References:**
>
> [1] Sébastien Bubeck. Convex Optimization: Algorithms and Complexity, 2015.
>
> [2] Yurii Nesterov. Lectures on Convex Optimization (Second Edition).
>
> ---
>
> **Additional explanation on universality.** We sincerely appreciate your attention to the convex and strongly convex cases in our paper (i.e., Thms 1-4). We would also like to highlight our progress in the universal setup (Thm 5), where we achieve the optimal universal rates with a simple OGD with one gradient query per iteration, a result not attainable by prior work.
>
> Notably, **the contribution in the universal setting builds directly upon our initial O2B conversion developed in Thm 1**. This progression, from the foundational convex case to the challenging universal setup, **demonstrates the effectiveness of our proposed framework of optimistic O2B conversions.** Therefore, the techniques presented across Thms 1, 3, and 5 collectively constitute our primary contribution to online-to-batch conversion methods.
>
> ---
>
> We believe that our work makes valuable contributions to the community. If our responses have adequately addressed your concerns about the contributions of our work, we kindly request a reevaluation of our paper's score. We are happy to provide further clarifications if needed during the author-reviewer discussions.

---

> > ### Comment · Reviewer_h8Yk · 2025-08-05
> >
> > Thank you. While the many relationships to [WA'18] limits the significant of this work, the universality result still can be of value to the community, and I raise my score to a limited yet positive support of this paper.
> >
> > I have no further questions.

---

> > > ### Author Response · Authors · 2025-08-06
> > >
> > > Thank you for the update and for re-evaluating our manuscript. We are very grateful for your time and for your final positive support. We are glad you found value in the universality result, and we appreciate all of your feedback, which has helped us improve the paper.

---

### Official Review · Reviewer_LDsK · 2025-06-24

**Clarity:** 4
**Significance:** 2
**Originality:** 2
**Rating:** 5
**Confidence:** 2

**Summary:**

This paper provides an overview and some new theory for online-to-batch conversions with a focus on universal and accelerated approaches.

**Questions:**

See Above

**Ethical Concerns:**

["NO or VERY MINOR ethics concerns only"]

**Final Justification:**

I'm generally in agreement with the other positive scoring reviewers, the relation to existing work is ok since this work generalizes the existing work.

**Limitations:**

See Above

**Quality:**

3

**Strengths And Weaknesses:**

Online-to-batch conversions are not yet well understood in the optimization community and I think this paper does a good job of introducing existing approaches and expanding on them.

- The proof techniques for the theory are a different and interesting take compared to the other approaches in the existing literature. Much of the overview content will be already familiar to researchers in the area, while for others it will be a good introduction and it's very useful to bring all these results together and provide a clarifying view.

- The exact correspondence between certain instantiations of optimism methods and Nesterov's acceleration is generally known to researchers in the area but is not clearly indicated in the literature, so the discussion here is welcome.

- The use of the average iterate as the location of gradient evaluation was also separately discovered in the stochastic optimization literature separately from the online learning community, under the name “Primal Averaging” by Tao et al. [2020], and was further explored in “The Power of Factorial Powers” (Defazio and Gower, 2020). Some discussion of this related work would be welcome. A variant of these  online-to-batch conversions were also very recently shown to actually be usable for real-world deep learning training in "The Road Less Scheduled".

- The UniXGrad paper (cited briefly already) also uses a similar difference-of-gradient's to define a universal method. Is there any relation between the methods? the UniXGrad paper uses general divergences which makes it not immediately clear to me.

---

> ### Author Rebuttal · Authors · 2025-07-30
>
> Thank you for the positive feedback and appreciation of our work! We answer your questions in the following.
>
> ---
>
> **Q1.** The use of the average iterate as the location of gradient evaluation was also separately discovered in the stochastic optimization literature separately from the online learning community, under the name "Primal Averaging" by Tao et al. [2020], and was further explored in "The Power of Factorial Powers" (Defazio and Gower, 2020). Some discussion of this related work would be welcome. A variant of these online-to-batch conversions were also very recently shown to actually be usable for real-world deep learning training in "The Road Less Scheduled".
>
> **A1.** Thank you for the suggestion. We will discuss the works about the use of the average iterate as the location of gradient evaluation in the revised version to provide readers a more comprehensive review of the literature. We will also discuss the paper of "The Road Less Scheduled" about its important application to real-world deep learning training.
>
> ---
>
> **Q2.** The UniXGrad paper (cited briefly already) also uses a similar difference-of-gradient's to define a universal method. Is there any relation between the methods? the UniXGrad paper uses general divergences which makes it not immediately clear to me.
>
> **A2.** Thank you for the question. Both UniXGrad and our Sec 3.3 aim to design a universal method in convex smooth optimization. The difference is that UniXGrad uses $\alpha_t \nabla f(\bar{x}_t)$ as the gradient and $\alpha_t \nabla f(\tilde{x}_t)$ as the optimism (their mirror-prox method can be seen as a variant of optimistic online learning algorithm), while our algorithm requires **only one gradient** of $\alpha_t \nabla f(\tilde{x}_t)$ and **no optimism**, because **our optimistic online-to-batch conversion has already incorporated optimism directly in the analysis**. We will revise the paper to explain the difference between our method and UniXGrad more clearly.
>
> ---
>
> We will carefully revise the paper and add the related works according to your suggestions. Thank you!

---

> > ### Comment · Reviewer_LDsK · 2025-08-07
> >
> > Thanks for the response. I'm generally in agreement with the other positive scoring reviewers, the relation to existing work is ok since this work generalizes the existing work. I will keep my positive score.

---

> > > ### Author Response · Authors · 2025-08-08
> > >
> > > Thank you for confirming your continued support for our paper. We are grateful for your perspective on how our framework generalizes existing work. We truly appreciate your positive assessment.

---

### Official Review · Reviewer_Wpuw · 2025-07-02

**Clarity:** 3
**Significance:** 1
**Originality:** 1
**Rating:** 4
**Confidence:** 4

**Summary:**

This paper studies an online to batch conversion approach to convex optimization. The primary contribution of this work is demonstrating that online algorithms with optimistic hints are not necessary to obtain accelerated rates. The key change to the algorithm is evaluating the update gradient at a point which approximates the average of the iterates, without requiring access to (yet to be determined) next iterate. Due to the clever choice of evaluation point, a simple online to batch analysis results in an optimal rate with no requirement for direct hinting. The method can be extended to utilize adaptive step sizes in a manner robust to smoothness assumptions.

**Questions:**

- My reading is that Theorem 4 implicitly requires unconstrained problem to ensure $ x_t $ is feasible and $ \nabla f( x^* ) = 0 $ for the final inequality in proof. Is this the case? If so you should make this explicit in the statement of the the theorem. Also the proof could be tightened by using the fact that $|| \nabla f(\tilde{x}_1) || = || \nabla f(\tilde{x}_1)- \nabla f( x^* )|| \leq L|| \tilde{x}_1 - x^* ||$.
- Could you discuss why the change in the algorithm is necessary in Theorem 4?
- You should make the $t$ values (e.g., $t\geq0$?) that (16)-(17) are applicable explicit and prove the edge cases hold (particularly for (17) which includes the $x_{t-2}$ term). Similarly, you should define the edge cases for $\tilde{x}_0$, $\bar{x}_0$, and $A_0$ etc.
- Could you elaborate on how (12) is obtained? What should $\beta_{t-1}$ be?
- line 566 why is the expectation conditioned on $x$?
- I think that Section 1.3 could benefit from being cut down into a few dot points which highlight the main contributions.
- Could you explain the working after line 594?

Typos:
- line 162: $\ell_t$ is a scalar, so this statement doesn't make sense.
- line 346: worth should be worthy?
- In the working after line 476 I think the final equality should be an inequality.
- line 527: "the second step uses Lemma 3" should this be Lemma 2?
- line 557: should $2G^2$ be $4G^2$ after the last inequality?
- line 585: filtration should be adapted to $\tilde{x}_i$s instead of $x_i$s?

**Ethical Concerns:**

["NO or VERY MINOR ethics concerns only"]

**Final Justification:**

As noted in my original review, my primary concern with this paper is the insufficiently acknowledged similarity to [WA18], including an identical algorithm and closely related analysis in Theorems 1–4. Over the course of the rebuttal, the authors acknowledged the need to revise their manuscript to more clearly reflect this connection and to reframe their contribution as an O2B-based analysis of an existing algorithm. That said, the paper does offer a nontrivial extension—specifically, the development of an adaptive step size scheme that avoids additional gradient evaluations, as presented in Theorems 5 and 6. With the substantial revisions outlined in the rebuttal, I would lean towards accepting the paper. However, since reviewers will not have the opportunity to evaluate the revised manuscript (including the promised numerical results), I believe the paper remains at best borderline at this stage. Ultimately, the decision hinges on whether the AC and reviewers trust the authors to carry out the proposed changes effectively.

**Limitations:**

See strengths/weaknesses.

**Paper Formatting Concerns:**

The large case NAG, GD, HB etc. doesn't look great and is a bit distracting. Could this be changed to just bold, e.g., **NAG**?

**Quality:**

2

**Strengths And Weaknesses:**

### Strengths
- The explanation of the mechanism of the update mechanism, in comparison to previous anytime online to batch conversion is insightful and intuitive. Generally I think the discussion of previous anytime online to batch algorithms is good.
- Most of the mathematical exposition is clear and correct (minus a few technicalities; see questions). Most of thee proofs are relatively easy to follow.

### Weaknesses

**Insufficient Differentiation/Discussion of Previous work**
My primary concern is that I am unsure whether this paper is a sufficiently novel/interesting contribution to the community. In particular, I don't think that you sufficiently highlight the connection to Wang and Abernethy (2018) on the following fronts:
- Algorithm 2 in their paper looks to be essentially identical to your update. This fact should be highlighted in your remark/previous work
- The analysis leading to your Theorem 2 is very similar to their Theorem 2 and Corollary 1.
- Theorem 4 in your paper also appears to borrow heavily from their paper.
- Since your step is the same as Algorithm 2 in their paper and Proposition 1 is lifted directly, the results from Section 4 don't feel that interesting. They are more like previous work. Furthermore, I don't think you can claim that you have established the equivalence between your method and NAG as you do in line 100.

Overall I don't believe that the connection to Wang and Abernethy (2018) is sufficiently explained by the paper (in particular Remark 1 is certainly insufficient) which leaves me doubtful of exactly where the boundary between this contribution of this work and previous work is. Note that I don't have an issue building off of previous work per se, but I don't think this paper accurately summarises the relationship to previous work. Could you outline specifically how your method can be differentiated from this previous work/what your specific contribution is? So far as I can tell the adaptive step size is new but I am unsure how strong this result is.

**Stochastic Case**
Other works analyze the stochastic case in conjunction with the deterministic case, e.g, Cutosky (2019), Kavis (2019) and Joulani et al. (2022b). In particular, expectation bounds (under unbiased and bounded variance gradients assumptions) with some dependence on $\sigma$ usually arise naturally from the online to batch framework. What were the barriers to taking this approach for, say, theorem 1-4? The stochastic analysis in this paper is focussed on one specific case and an almost sure boundedness (which is stronger than a bounded variance assumption).

**Numerical Evaluation**
A numerical comparison could significantly elevate the contribution of this paper. In particular, I would be interested to see if there is an advantage to integrating the optimism into the evaluation point, rather than though a hinting mechanism (even if there is a negative result).

**Conclusion**
Overall I am leaning towards a borderline reject pending your response to the weaknesses/technical questions. I could move to accept/reject depending on your responses. I am particularly interested if you can clearly explain 1. Your contribution over and above the results from Wang and Abernethy (2018) and 2. How you will adjust your introduction/summary of previous work to highlight how connected your method is with Wang and Abernethy (2018).

---

> ### Author Rebuttal · Authors · 2025-07-30
>
> Thanks for the valuable feedback and very careful check of our paper! Below, we restate the contributions of our paper and the major differences from Wang and Abernethy (2018) [WA'18], hoping to resolve your concerns.
>
> ---
>
> **W1 & W2. About the connection/difference to [WA'18] and our own contributions.**
>
> * Algo 2 in their paper looks to be essentially identical to your update. The analysis leading to your Thm 2 is very similar to their Thm 2 and Cor 1.
> * Thm 4 in your paper also appears to borrow heavily from their paper.
> * The results from Sec 4 are more like previous work. Furthermore, I don't think you can claim that you have established the equivalence between your method and NAG as you do.
>
> **A1.** Thank you for your comments and suggestions. Below, we clarify the connections/differences with [WA'18] and explain our contributions beyond their work.
>
> * **Clarifications of the connection to [WA'18]:** First, we clarify that the convex optimization can be solved via an *online-to-batch (O2B) conversion* and an *online algorithm*. The design and analysis of online algorithms *heavily* depend on the O2B conversions. In our paper, **Thms 1,3,5 are our proposed optimistic O2B conversions** and **Thms 2,4,6 are the corresponding algorithmical analyses**. In this sense, **Thms 2,4,6 are actually corollaries of Thms 1,3,5**, and **the reviewer might mistake Thms 2,4,6 as our main contributions**. We will use "corollary" to state the current Thms 2, 4, 6 to clarify our own contributions to the reader. Below, we provide one-by-one clarifications.
>     * **Thm 2 is a corollary of Thm 1**. In Thm 1, our work and [WA'18] obtain the same intermediate result of $A_T [f(\bar{x}\_T) - f(x^\*)] \le \sum_{t=1}^T \langle \alpha_t \nabla f(\tilde{x}\_t), x_t - x^* \rangle + \sum_{t=1}^T \alpha_t \langle \nabla f(\bar{x}\_t) - \nabla f(\tilde{x}\_t), x_t - x_{t-1} \rangle$.  As explained in Line 179, the first term of the RHS is a look-ahead online problem, and solving it only requires the *standard* OGD with *common* analyses, making our Thm 2 and the proof similar to theirs. **We will add a remark to explicitly state that our Thm 1 obtains the same intermediate result as [WA'18], but from a completely different perspective, in the next version.**
>     * **Thm 4 is a corollary of Thm 3**. Both our work and [WA'18] aim to handle an online learning problem with regret: $\sum_{t=1}^T \alpha_t[h_t(x_t)-h_t(x^*)]$, where $h_t(\cdot)$ is a strongly convex surrogate loss. Therefore, solving this *standard* online problem leads to similar analyses. **We will add a remark to state the analytical similarity of Thm 4 to [WA'18].**
>     * Thanks for your advice. The statement of "we establish the equivalence between our method and the classical NAG" in Line 100 is imprecise, and **we will revise it to give correct credits to [WA'18] in the next version.** Note that in Prop 1, we have given the proper credits by citing their Thm 4. Our intention was not to claim this connection as our contribution, but to highlight this interesting connection between recent O2B-based methods and classical ones such as NAG and HB for readers.
>
> * **Differences from [WA'18]:**
>     * **Techniques behind our Thm 1 and Thm 3 are entirely new and different from [WA'18]**, as explained in Remark 1. They modeled the convex optimization problem as a two-player Fenchel game using the definition of Fenchel conjugate, while we start from the line of stabilized O2B conversion initiated by Cutkosky (2019).
>     * Furthermore, we aim to design a **universal-to-smoothness** method that does not know whether the objective is smooth or Lipschitz and can adapt to both cases automatically. This is **not** studied by [WA'18]. For this problem, we propose **a novel optimistic O2B conversion (in Thm 5)**, which allows **a single-step OGD with one gradient query** per iteration with the optimal universal convergence, which **previous works cannot do**.
>
> In general, we **make a step forward in the field of O2B conversions by proposing the more powerful optimistic O2B conversions (in Thm 1,3,5)**, which help achieve the optimal rates in (strongly) convex optimization and the universal setup with simple online algorithms.
>
> ---
>
> **W3.** About the stochastic cases.
>
> - Other works analyze the stochastic case in conjunction with the deterministic case, e.g, Cutkosky (2019), Kavis et al. (2019), and Joulani et al. (2022b). The stochastic analysis in this paper is focused on one specific case and an almost sure boundedness (stronger than a bounded variance assumption).
> - What were the barriers to taking this approach for Thms 1-4?
>
> **A3.** Thanks for the comments.
>
> - For a **bounded** domain, a common assumption required by previous works (e.g, Cutkosky, Kavis et al., and Joulani et al.), extending to the stochastic case is **straightforward**. Therefore, we defer it to the appendices due to page limits. Besides, we assume an almost sure boundedness because we aim to achieve **high-probability** rates (please kindly refer to our Thm 7), which is **stronger** than the *expectation* rates obtained by previous works, where a bounded variance assumption suffices.
> - On the other hand, handling stochastic noises in *unbounded* domains is challenging. Because our Theorem 1-4 holds even in *unbounded* domains, extending them to the stochastic setup is beyond the scope of this work, and we leave them as an interesting future direction. Thanks for the question.
>
> **W4.** Numerical Evaluation. In particular, I would be interested to see if there is an advantage to integrating the optimism into the evaluation point, rather than through a hinting mechanism (even if there is a negative result).
>
> **A4.** Thank you for the question. We agree that a numerical comparison would further strengthen the paper. As PDF attachments and external links are not permitted in this year's rebuttal, **we commit to adding experiments** comparing our algorithm with hinting-based methods (e.g., Cutkosky, 2019; Kavis et al., 2019) in the camera-ready version.
>
> ---
>
> **Q1.** Thm 4 implicitly requires an unconstrained problem to ensure $x_t$ is feasible and $\nabla f(x^*)=0$ for the final inequality in the proof.
>
> **A5.** Thank you for the sharp observation. You are right that Thm 4 indeed requires an unbounded domain. As explained in Line 128, the boundedness is only required for our universal method in Sec 3.3. And in Thm 4, we only require a smoothness assumption and do **not** import the boundedness assumption. **We will emphasize this independence in the revised version.**
>
> **Q2.** Could you discuss why the change in the algorithm is necessary in Thm 4?
>
> **A6.** Thanks for the question. This is a **necessary** adaptation to handle the geometry of strongly convex problems. The change allows the algorithm to leverage strong convexity to achieve a faster, linear convergence rate, which is analogous to how online learning algorithms are modified when moving from convex to strongly convex settings.
>
> **Q3.** You should make the $t$ values (e.g., $t \ge 0$?) that (16)-(17) are applicable explicit and prove the edge cases hold (particularly for (17), which includes $x_{t-2}$). Similarly, you should define the edge cases for $\tilde{x}_0$, $\bar{x}_0$, and $A_0$ etc.
>
> **A7.** Thank you for the advice to make our analysis more rigorous. We will add the edge definitions of $A_0 = 0$ and $\tilde{x}_1 = x_0$ such that: (14), (15), (16) hold when $t=1$; and (17) holds from $t=2$ which we indeed use from $t=2$, as in Eq. (21). **We will add the proof of the edge cases in the next version**.
>
> **Q4.** Could you elaborate on how (12) is obtained? What should $\beta_{t-1}$ be?
>
> **A8.** Thank you for the question. The proof of (12) follows a similar argument to Eq. (25) in the proof of Prop 1 by replacing $x_t = x_{t-1} - \eta_{t-1} g_{t-1}$ into (25). Therefore, $\beta_{t-1} = \frac{\alpha_t A_{t-2}}{A_t \alpha_{t-1}}$ in (12), which is the same as that in (25). We **will add a corresponding proof in the appendix in the next version.**
>
> **Q5.** Line 566 why is the expectation conditioned on $x$?
>
> **A9.** Thank you for the question. This is a standard assumption in stochastic optimization and shows that the unbiasedness does *not* depend on what the current decision $x$ is, which is reasonable and is a fundamental learnable assumption in stochastic optimization.
>
> **Q6.** Sec 1.3 could benefit from being cut down into a few dot points that highlight the main contributions.
>
> **A10.** Thanks for the suggestion. **We will restructure Sec 1.3 into a list** to highlight our main contributions more clearly.
>
> **Q7.** Could you explain the working after line 594?
>
> **A11.** We are not sure if we understand your question concretely. As we aim to prove our algorithm is *universal to smoothness*, we discuss two cases separately after Line 594. In the smoothness case, the main proof technique is the self-confident tuning, a clever idea from online learning. In the non-smoothness case, we directly bound the term using the assumption of $\\|g(x)-\nabla f(x)\\| \le \sigma$. If our answer does not solve your question, we welcome further discussions in the upcoming discussion period.
>
> **Q8.** About Typos and Format.
>
> **A12.** Thank you for your careful reading. We will correct all the noted typos and formatting issues in the revised version. For brevity, we are not listing them one by one here due to the character limit.
>
> ---
>
> We believe that our work offers valuable contributions to the community and will carefully revise the paper according to your suggestions and questions. If our responses have adequately addressed your concerns about the contributions of our work, we kindly request a reevaluation of our paper's score. And we are happy to provide further clarifications if needed during the following author-reviewer discussions.

---

> > ### Comment · Reviewer_Wpuw · 2025-08-03
> >
> > Thank you for your detailed response to my review. After reading your responses and the other reviews there are just a few points I would like to clarify.
> >
> > **A1** From reading your responses, going back over the paper and [WA18] how your approach does differ in motivation (in particular, by considering the stabilized O2B setting). I think there is certainly merit to the reframing to an O2B setting. Indeed, this is clearly illustrated by your results in Theorem 5 and 6. However, one issue that has not been addressed, so far, is that so far as I can tell Algorithm 2 in [WA18] is the same as your Algorithm 2. I feel like this is the fundamental reason the proof techniques for Theorem 1-4 are very similar (despite differing motivations). Could you comment on this?
> >
> > I want to stress that I think it is ok to overlap with other work, including giving a fresh perspective on a known algorithm. However, at the moment I think the narrative of the paper requires reworking to acknowledge that this is the approach being taken.
> >
> > **A4** I think this would be a great comparison!
> >
> > **A6** "which is analogous to how online learning algorithms are modified when moving from convex to strongly convex settings" it could be helpful to include an example to contextualize this change of setting for Theorem 4. At the moment the switch of settings/algorithm feel abrupt.
> >
> > I think the other changes you list will certainly be positive updates for the manuscript.
> >
> > A few additional typos
> > - In line 472-473 $\eta = 1/L$ should be $\eta = 1/(4L)$?
> > - After line 484 should be $REG_T \leq \ldots$ instead of equality?

---

> ### Author Response · Authors · 2025-08-04
> **Response I: Clarification on Framework and Contribution**
>
> Thank you very much for the detailed follow-up responses and for acknowledging our contributions. Due to the character limit, we present our full responses in two separate posts, where we clarify our main contribution in relation to [WA'18] in the first response and address your other valuable suggestions in the second one.
>
> ---
>
> **Q1.** From reading your responses, going back over the paper and [WA18] how your approach does differ in motivation (in particular, by considering the stabilized O2B setting). I think there is certainly merit to the reframing to an O2B setting. Indeed, this is clearly illustrated by your results in Thms 5 and 6. However, one issue that has not been addressed, so far, is that so far as I can tell Algo 2 in [WA18] is the same as your Algo 2. I feel like this is the fundamental reason the proof techniques for Thms 1-4 are very similar (despite differing motivations). Could you comment on this?
>
> **A1.** Thank you for this insightful question and for the detailed reading. You are correct about **Algo 2 is indeed identical to Algo 2 in [WA'18]**, and we sincerely appreciate you pointing this out.
>
> You also raised a sharp point about this being the **fundamental reason** for the similar proof techniques. From our perspective, this similarity arises because both our O2B framework and their Fenchel game formulation are successful in reducing the optimization problem down to the **same underlying online learning subproblem** (i.e., minimizing the regret $ \sum_{t=1}^T \langle \alpha_t \nabla f(\tilde{x}\_t), x_t - x^* \rangle$). Once this shared fundamental structure is revealed, the standard OGD becomes the most **natural** algorithmic choice, and its analysis inevitably follows a similar and **standard** path. Therefore, the identical algorithms and similar proofs are a **necessary consequence of both frameworks identifying the same core challenge**.
>
> Furthermore, we agree the paper's current narrative needs to be reworked to reflect this. Your suggestion to frame our contribution as **"giving a fresh perspective on a known algorithm"** is precisely the right approach, and we are grateful for this constructive guidance. In the revised version, we have reshaped the paper's narrative as follows:
>
> 1. **In the Introduction**, we have explicitly stated that our work **provides a new and unifying O2B framework**. We have clarified that, as **a first application**, our framework can be used to **re-derive** the algorithm from [WA'18] (our Algo 2), **offering new insights into why it works**.
> 2. **When introducing Algo 2,** we have clearly stated that it is **the same as** Algo 2 in [WA'18] and that our contribution lies in demonstrating **how** our O2B conversion naturally leads to this effective algorithm.
> 3. This serves as a **bridge** to then highlight our **main novel contributions in Thms 5 and 6**, showing that our framework is not only powerful enough to explain existing results but also to generate entirely new ones that were not attainable by prior approaches.
>
> We believe this revised narrative, thanks to your suggestion, will present our contributions more accurately and transparently. Thank you again for helping us significantly improve the paper.
>
> ---
>
> Our responses to your other points (**Q2-Q4**) follow in the next post.

---

> ### Author Response · Authors · 2025-08-04
> **Response II: Clarification on Other Suggestions**
>
> In this response, we address your remaining suggestions on the numerical evaluation, a clarifying example, and typos.
>
> ---
>
> **Q2.** About the numerical evaluation. "I think this would be a great comparison!"
>
> **A2.** We agree that comparing our methods with the hinting-based methods (e.g., Cutkosky, 2019; Kavis et al., 2019) would greatly validate their effectiveness. We commit to including these comparisons in the camera-ready version of the manuscript as an additional page is allowed.
>
> ---
>
> **Q3.** "which is analogous to how online learning algorithms are modified when moving from convex to strongly convex settings" it could be helpful to include an example to contextualize this change of setting for Thm 4.
>
> **A3.** Thank you for your suggestion. In the revised version, we have added a remark to provide context for this algorithmic change. We use a classic, well-known example from online convex optimization (OCO) to illustrate why different function geometries require tailored algorithms.
>
> > **Remark 3 (Context for the Algorithmic Change).** In Thm 4, a different algorithm is required to leverage the geometry of strongly convex problems for faster rates. This phenomenon is standard in optimization theory. A classic analogy is found in online convex optimization (OCO), where the standard Online Gradient Descent (OGD) algorithm is modified based on the function's curvature. For general convex functions, OGD typically uses a learning rate of $\eta_t \propto 1/\sqrt{t}$ to achieve the optimal $O(\sqrt{T})$ regret [1]. However, for $\lambda$-strongly convex functions, the algorithm is adjusted to use a more aggressive learning rate of $\eta_t \propto 1/(\lambda t)$ to exploit the strong convexity and achieve a much faster $O(\log T)$ regret [2]. For another widely-studied function class called exp-concavity, a fundamentally different algorithm is required for the optimal regret [2]. These examples illustrate the principle that adapting the algorithm to the problem's structure is crucial for achieving optimal performance.
>
> **References:**
>
> [1] Online Convex Programming and Generalized Infinitesimal Gradient Ascent, 2003
>
> [2] Logarithmic Regret Algorithms for Online Convex Optimization, 2007
>
> ---
>
> **Q4.** A few additional typos.
>
> **A4.** Thank you for your careful reading! Below, we clarify the typos in your first review (omitted due to the character limits of the rebuttal) and then the typos in this review.
>
> * **Typos - I:** In Line 162, it should be $\\|\nabla \ell_t(w_t)-M_t\\|^2$ but not $\\|\ell_t(w_t)-M_t\\|^2$. In Line 346, it should be "worthy" but not "worth". In Line 476, it should be "$\le$" but not "$=$". In Line 527, it should be "Lemma 2" but not "Lemma 3". In Line 557, it should be "$4G^2$" but not "$2G^2$". In Line 585, you are correct, the filtration should depend on $g(\tilde{x}_t)$ but not $g(x_t)$.
> * **Typos - II:** In Line 172, it should be $\eta = 1/(4L)$ but not $\eta = 1/L$. In Line 484, it should be $\text{Reg}_T \le \cdots$ but not $\text{Reg}_T = \cdots$.
>
> We have corrected all the above typos in the revised version and thank you again for your suggestions for improving this paper!
>
> ---
>
> We believe that our work makes valuable contributions to the community. If our responses have adequately addressed your concerns about the contributions of our work, we kindly request a reevaluation of our paper's score. We are happy to provide further clarifications if needed during the author-reviewer discussions.

---

> > ### Comment · Reviewer_Wpuw · 2025-08-04
> >
> > Thank you for your detailed responses, I think you have clarified my remaining concerns.

---

> > > ### Author Response · Authors · 2025-08-06
> > >
> > > Thank you for your feedback and for confirming that we have addressed your concerns. We sincerely appreciate your time and your insightful comments, which have helped us improve the manuscript significantly.

---

### Official Review · Reviewer_zkAz · 2025-07-02

**Clarity:** 4
**Significance:** 3
**Originality:** 2
**Rating:** 5
**Confidence:** 3

**Summary:**

This paper proposes a new online-to-batch conversion method and analysis technique to achieve acceleration from online gradient descent with gradients weighted differently. In contrast, previous online-to-batch conversion achieved accelerated rates when using some *optimistic* online learning algorithms. The idea of the analysis is to, in some sense, incorporate optimism only in the analysis, not as a requirement on the OL method itself. With that they show accelerated rates in the smooth and smooth strongly convex cases. Moreover, using their online-to-batch techniques they also show how to get acceleration without knowing the smoothness constant beforehand when the domain is bounded, what is known as an *universal algorithm*.

**Questions:**

My main question is whether Algorithm 2 (and Theorem 5) depend on the use of online gradient descent, of whether we could use (and if it makes sense) to use other online learning algorithms.

**Ethical Concerns:**

["NO or VERY MINOR ethics concerns only"]

**Final Justification:**

I might still edit this if the discussion with the authors develops further. Currently it seems like the main contribution of this paper is a new perspective (optimistic O2B) to obtain known results and algorithms. Some of the convergence proofs at some points are very close to those of WA18, but it seems, to me at this point, quite non-obvious how this work gets to the same "absract bounds on $\alpha$-regret" as WA18 starting from their framework. Since O2B reductions are quite important in OL, I think that this new perspective on known results is going to be of interest to the community.

**Limitations:**

Yes.

**Paper Formatting Concerns:**

No formatting concerns

**Quality:**

4

**Strengths And Weaknesses:**

I believe this is a very strong submission: clearly written, with an elegant main idea and with very nice results. Of course, I may be missing some of the context from the recent work on online-to-batch conversion that happened more recently, but this feels like a simple method with a clean analysis and very strong results. I really appreciated the thorough discussion that the authors do with respect to related work. I was familiar with the use of the Fenchel game from Abernethy et al., so the connection to this work was quite interesting to me. Beyond acceleration in both smooth and smooth + stongly convex case, the authors also show how to get acceleration in bounded domains without know the smoothness constants, which was a very neat addition. I did not have the time to check the proofs for correctness properly, but the ideas in this paper are quite interesting.

The one main thing I was a bit confused was whether this optimistic online-to-batch conversion is restricted to online gradient descent. Algorithm 1 is describe with a general OL method and I know some of the results end up using O-t-B conversion with other algorithms, but in the paper the optimistic O-t-B conversion is described explicitly using OGD. I did not have the time to check whether Theorem 5 actually depends on the algorithm being OGD and I feel it was a bit confusing how dependent on it this method is.

---

> ### Author Rebuttal · Authors · 2025-07-30
>
> Thank you for your strong support for our paper! In the following, we answer your question about whether online gradient descent is coupled with our optimistic online-to-batch conversion.
>
> ---
>
> **Q.** My main question is whether Algo 2 (and Thm 5) depend on the use of online gradient descent, of whether we could use (and if it makes sense) to use other online learning algorithms.
>
> **A.** Thank you for the insightful question. The choice of online learning algorithms is **flexible**. We chose to present our main results with Online Gradient Descent (OGD) primarily to highlight the power of our optimistic conversion framework: **it simplifies the underlying task to a degree where even a standard algorithm like OGD is sufficient to achieve optimal rates**.
>
> In the future, when handling more challenging tasks, such as *parameter-free accelerated optimization*, as discussed in Line 298, traditional conversion requires the online algorithm to achieve both parameter-freeness and optimism. On the other hand, since our conversion incorporates optimism intrinsically, we only require the online algorithm to enjoy parameter-freeness. This could reduce the burden of online algorithm design, which might make such advancements possible. We leave this as a promising future direction.
>
> ---
>
> Once again, we are very grateful for your positive feedback and valuable encouragement. We would be happy to provide any further clarifications needed.

---

> > ### Comment · Reviewer_zkAz · 2025-08-04
> >
> > I would like to thank the authors for their replies to my reviews and also for engaging in very nice discussion with other reviewers. I really think the review from Wpuw was extremely good and very important to make this manuscript better in the end. I wasn't aware that the similarities between this work and WA18 were so strong. (And I was definitely over-confident on my original review).
> >
> > The proof of Thm 2 of the current submission is definitely very similar to the one of WA18. Like the authors also mention in the rebuttal to Wpuw, the main point of difference is how this submisson and WA18 arrives at (the equivalent of) Thm 1. This is fine, but I strongly agree with Wpuw that **these similarities should be more clearly discussed in this paper**. I still believe that the proposed optimistic O2B perspective is insightful, but it would be of benefit both for readers and for the strength of the submission to make these similarities much clearer.
> >
> > At this point, it is unclear to me whether one could achieve acceleration without knowledge of L using the Fenchel game perspective (and here I am considering the more general view from [1], which I believe the authors should likely cite as related work). Are the authors aware whether it is possible to achieve similar results to the ones of Theorem 5 or 6 using the Fenchel game framework? I do not think this would be a reason to reject the current submission, but I think that, given the similarities, it would be *very* interesting to know if one cat get these results via the Fenchel game. The authors should not feel that they need to reply to this in such a short notice, but if the authors knows a way to get these results via the Fenchel game, I would be interested.
> >
> > In the end, it might even be that these two approaches are more deeply connected (that is, this O2B can be obtained from the Fenchel game with a specific choice of $y$ player), but it seems to me that this would be a very non-trivial reduction and another project in itself.

---

> > > ### Author Response · Authors · 2025-08-06
> > >
> > > Thank you for your constructive follow-up. We strongly agree that the discussions, particularly with Reviewer #Wpuw, are very fruitful in helping us frame our contributions more precisely. Below, we provide the specific revisions regarding the connection to [WA'18] and share our thoughts on your question about the universality of the Fenchel game framework.
> > >
> > > ---
> > >
> > > **Q1.** The proof of Thm 2 of the current submission is definitely very similar to the one of WA18. I strongly agree with Wpuw that these similarities should be more clearly discussed in this paper.
> > >
> > > **A1.** Thank you for this crucial suggestion. We agree completely that a clearer discussion of these similarities strengthens the paper. To that end, we have incorporated the following remarks into our revision:
> > >
> > > * In the Introduction (Lines 78-81), we have added the following remark:
> > >
> > >     > Our work is inspired by Wang and Abernethy [2018] and arrives at the same intermediate result (in Theorem 1) through our novel optimistic online-to-batch conversion framework. As a first application, our framework re-derives the algorithm from [WA'18] (our Algorithm 2), offering new insights for validating its effectiveness.
> > >
> > > * In Section 3.1 (after Thm 2), we have added the following remark:
> > >
> > >     > We note that Algorithm 2 is identical to that in Wang and Abernethy [2018], a consequence of both works identifying the same underlying online learning subproblem (Theorem 1). Our contribution here is not a new algorithm, but rather a demonstration of our online-to-batch framework's power to provide a new pathway to this effective algorithm and, more importantly, to unlock new results later on (in Theorem 5 and 6) that were previously unattainable.
> > >
> > > ---
> > >
> > > **Q2.** Are the authors aware whether it is possible to achieve similar results to the ones of Theorem 5 or 6 using the Fenchel game framework?
> > >
> > > **A2.** Thank you for the insightful question! Given that our work and [WA'18] achieved the same intermediate result shown in Thm 1, [WA'18] can be easily extended to the unversal setup by using an AdaGrad-type step size $\eta_t = D / \sqrt{\sum_{s=1}^t \alpha_s^2 \\| \nabla f(\bar{x}_s) - \nabla f(\tilde{x}_s) \\|^2}$, as shown in Corollary 5. Note that this step size requires **two gradient queries per iteration**.
> > >
> > > To achieve **one** gradient query per iteration as Thm 5 and 6, our preliminary analysis suggests that achieving the single-gradient performance within the Fenchel game framework would be non-trivial. Two possible directions would be: (i) providing a new analysis of Lemma 2 for Optimistic FTL in [WA'18], the current algorithm of player-$y$, to achieve a regret guarantee depending solely on $\nabla f(\tilde{x}_t)$ without $\nabla f(\bar{x}_t)$; or (ii) designing a new online algorithm for player-$y$. We are working on both directions and will synchronize the most recent progress if discussions are still permitted.
> > >
> > > ---
> > >
> > > **Q3.** It might even be that these two approaches are more deeply connected (that is, this O2B can be obtained from the Fenchel game with a specific choice of player), but it seems to me that this would be a very non-trivial reduction and another project in itself.
> > >
> > > **A3.** Thank you for the insightful comment. Currently, the inner connection between these two frameworks is not obvious enough. We agree that the Fenchel game formulation is a powerful framework and we will continue working on its deep connection with the online-to-batch conversion based framework for future exploration.
> > >
> > > ---
> > >
> > > Thank you again for your constructive comments and suggestions. We believe that our work offers valuable contributions to the convex optimization problem from the perspective of online-to-batch conversions. We will revise the paper throughoutly according to the comments from you and the other reviewers, especially Reviewer #Wpuw, to present the contributions of this manuscript more clearly and accurately.

---

### Official Review · Reviewer_SPDa · 2025-07-03

**Clarity:** 3
**Significance:** 3
**Originality:** 4
**Rating:** 5
**Confidence:** 3

**Summary:**

This paper proposes a new method called optimistic conversion to leverage online learning analysis for offline convex optimization. While several conversion techniques have been developed in the past, achieving an $O(1/T^2)$ convergence rate for offline smooth convex optimization typically required analysis of optimistic online methods. In contrast, the proposed optimistic conversion enables the derivation of an offline algorithm equivalent to Nesterov's accelerated gradient method using only a simple, non-optimistic online gradient descent, while still attaining the $O(1/T^2)$ convergence. In the strongly convex case, linear convergence can also be achieved. Moreover, this conversion facilitates the development of universal methods that do not rely on smoothness, as well as stochastic variants.

**Questions:**

Do the universal algorithms presented in Section 3.3 achieve linear convergence, similar to Theorem 3, when applied to strongly convex and smooth functions?

**Ethical Concerns:**

["NO or VERY MINOR ethics concerns only"]

**Final Justification:**

- The proposed optimistic conversion appears promising and helps simplify algorithm design.
- The authors responded clearly and adequately addressed the concern regarding universality for strongly convex objectives.

**Limitations:**

yes

**Quality:**

3

**Strengths And Weaknesses:**

**Strengths**
- The paper addresses online-to-batch conversion, a timely and actively studied topic in the field.
- The idea of optimistic conversion is intriguing and has the potential to aid the design of algorithms with improved theoretical guarantees.
- The paper provides a thorough review of prior work, making it a valuable introduction to online-to-batch conversion. It also clearly positions the current work within the context of existing literature and describes their relationships well.
- The paper is overall well-written and easy to follow.

**Weakness**
- While the paper presents a new framework for algorithm design and analysis, it does not appear to lead to new algorithms that improve upon existing theoretical results at this stage.

---

> ### Author Rebuttal · Authors · 2025-07-30
>
> We sincerely thank you for your positive feedback and for recognizing the value of our paper. We appreciate the opportunity to clarify the theoretical contribution of our work and to address your question regarding the universal algorithm.
>
> ---
>
> **W.** While the paper presents a new framework for algorithm design and analysis, it does not appear to lead to new algorithms that improve upon existing theoretical results at this stage.
>
> **A1.** Thank you for the comment. We agree with the reviewer's observation that our work does not improve upon the *known optimal convergence rates*. However, it does not mean that further exploration on this problem is unnecessary.
>
> The main contribution of our paper lies on a different, yet equally important, axis: **we propose powerful optimistic online-to-batch (O2B) conversions for designing and understanding optimal algorithms in convex optimization.** Our contribution is analogous to the long thread of research that continues to provide new perspectives on Nesterov's Accelerated Gradient (NAG) method, decades after its discovery.
>
> Specifically, our more powerful optimistic O2B conversions can **reduce the burden of algorithm design**. For example, in the convex smooth optimization, a simple OGD suffices to achieve the optimal rate. Besides, in the universal setup, our optimistic O2B conversion allows **a single-step OGD with one gradient query** in each iteration to achieve the optimal universal convergence, which **cannot** be done by previous works. Finally, as explained in Sec 4, the progress in this thread might also provide **new understandings of the classical NAG from the perspective of online learning**. To make this point clearer, we have revised the paper to better clarify our contributions in this regard.
>
> ---
>
> **Q.** Do the universal algorithms presented in Sec 3.3 achieve linear convergence, similar to Theorem 3, when applied to strongly convex and smooth functions?
>
> **A2.** Thank you for the insightful question! Formally, for $\lambda$-strongly convex objectives, a universal method is expected to achieve $O(\exp (-T / \sqrt{\kappa}))$ for smooth functions and $O(1/ [\lambda T])$ in the Lipschitz case, where $\kappa = L/\lambda$ represents the condition number. **How to achieve the optimal rates is still an open problem and cannot be easily solved via direct extensions of the universal method in Sec 3.3.** Specifically, in our method, the O2B conversion weight $\alpha_t$ is chosen as $\alpha_t = C A_{t-1}$, where $C$ is a constant depending on the condition number $\kappa$. Therefore, in the universal setup where the smoothness parameter $L$ is unknown, achieving universality is more challenging than the convex case because the method needs to estimate the smoothness parameter on the fly. There are some preliminary and sub-optimal results, e.g., the pioneering work of Levy (2017), where the author obtained a *non-accelerated* rate of $O(\exp (-T / \kappa) \cdot T / \kappa)$ for strongly convex smooth objectives and $O((\log T) / T)$ for strongly convex Lipschitz objectives. We have a separate paper under review that handles this problem, justifying its separate treatment due to its complexity. And we will add a remark in Sec 3.3 to clarify this issue in the revised version.
>
> ---
>
> We believe that our work offers valuable contributions to the community in the field of online-to-batch-conversion-based methods. We hope our responses have addressed your concerns and would appreciate a reevaluation of our paper's score. And we are happy to provide further clarifications if needed during the following author-reviewer discussions.

---

> > ### Comment · Reviewer_SPDa · 2025-08-05
> >
> > Thank you for the detailed and clear response.
> >
> > I found Answer 2 particularly satisfying, as it helped me understand that what I had considered a (minor) issue in the paper is actually an open problem, one so challenging that it deserves a dedicated paper. I have raised my score to 5.

---

> > > ### Author Response · Authors · 2025-08-06
> > >
> > > Thank you for the positive feedback and for your support in raising the score. We truly appreciate your time and engagement throughout the review process, which have been invaluable to us. We agree that achieving universality in strongly convex optimization is a very interesting problem, and it is a direction we will certainly explore in our future work, aiming to achieve optimal rates or design detection-free methods. We hope that our forthcoming work on this topic will be of interest to you.

---

### Note · Authors · 2025-08-15

We sincerely thank all reviewers and the AC for their time and constructive feedback. Based on these discussions, we have revised the manuscript to better position our work within the literature, particularly [WA’18], and highlight our distinct contributions.

---

**Main contributions:**

Our work provides several *optimistic online-to-batch (O2B) conversions* that simplify the design of optimal algorithms in convex optimization while preserving the optimal theoretical guarantees:

* For convex smooth optimization, we show that a single descent–descent step achieves the optimal convergence. Our O2B conversion re-derives the method in [WA’18], providing fresh perspectives on its effectiveness.
* For strongly convex smooth optimization, by applying the optimistic O2B conversion with online algorithms tailored for strong convexity, we achieve the optimal rate directly.
* For universal-to-smoothness optimization, our improved optimistic O2B conversion enables a single-step OGD with only one gradient query per iteration to achieve optimal universal convergence, which is unattainable by prior work.

**Discussions of [WA'18]:**

We have included the following clarifications in our revised manuscript:

> We note that Algorithm 2 is identical to that of [WA’18], as both methods ultimately reduce the problem to the same underlying online learning subproblem (Theorem 1). Our contribution, therefore, is not a new algorithm, but rather demonstrating how our O2B framework naturally leads to this known method, providing new understanding of its effectiveness, and more importantly, enables new results (Theorems 5 and 6) that were previously unattainable.

> While the framework of [WA’18] can also achieve universal convergence using an AdaGrad-type step size, this requires two gradient queries per iteration. Achieving optimal universal convergence with only one gradient query per iteration within their framework remains a challenging open problem.

---

We believe that our work makes a step forward in the study of online-to-batch conversion–based methods. We will incorporate the valuable suggestions from all reviewers—particularly those concerning [WA’18]—to further clarify, refine, and strengthen the manuscript. We will also continue investigating the open problems mentioned, such as achieving universal optimality in strongly convex optimization and addressing strongly convex stochastic optimization via online learning algorithms.

---

### Decision · Program_Chairs · 2025-09-17

**Decision:**

Accept (poster)

**Comment:**

This paper introduces optimistic online-to-batch conversions that incorporate optimism into the conversion step, allowing simple online gradient descent to achieve accelerated convergence. The framework also leads to a universal method that achieves optimal rates with only one gradient query per iteration, improving over prior work.

Reviewers appreciated the paper's clarity, clean analysis, and strong positioning within the literature. They highlighted the elegance of the optimistic conversion idea and the novel universality result, which goes beyond earlier work. While some overlap with Wang & Abernethy (2018) was noted, the authors clarified that these are corollaries of their framework, which provides a new perspective and enables the new contributions.

Overall, the paper is technically solid, well written, and offers both conceptual clarity and new results.